# CHARACTERISING PARTIAL IDENTIFIABILITY IN INVERSE REINFORCEMENT LEARNING FOR AGENTS WITH NON-EXPONENTIAL DISCOUNTING

## ABSTRACT

The aim of inverse reinforcement learning (IRL) is to infer an agent's *preferences* from their *behaviour*. Usually, preferences are modelled as a reward function, $R$, and behaviour is modelled as a policy, $\pi$. One of the central difficulties in IRL is that multiple preferences may lead to the same behaviour. That is, $R$ is typically underdetermined by $\pi$, which means that $R$ is only *partially identifiable*. Recent work has characterised the extent of this partial identifiability for different types of agents, including *optimal* agents and *Boltzmann-rational* agents. However, work so far has only considered agents that discount future reward exponentially. This is a serious limitation, for instance because extensive work in the behavioural sciences suggests that humans are better modelled as discounting *hyperbolically*. In this work, we characterise the partial identifiability in IRL for agents that use non-exponential discounting. Our results are relevant for agents that discount hyperbolically, but they also more generally apply to agents that use other types of discounting. We show that IRL, in these cases, is unable to infer enough information about $R$ to identify the correct optimal policy. This suggests that IRL alone is insufficient to adequately characterise the preferences of such agents.

## 1 INTRODUCTION

Inverse reinforcement learning (IRL) is a subfield of machine learning that aims to develop techniques for inferring an agent's *preferences* based on their *actions*. Preferences are typically modelled as a reward function, $R$, and behaviour is typically modelled as a policy, $\pi$. An IRL algorithm must additionally have a *behavioural model* that describes how $\pi$ is computed from $R$. By inverting this model, an IRL algorithm can then deduce $R$ from $\pi$. There are many motivations for IRL: for example, it can be used in imitation learning (e.g. Hussein et al., 2017), or it can be used as a tool for *preference elicitation* (e.g. Hadfield-Menell et al., 2016). In this paper, we are primarily concerned with IRL in the context of preference elicitation.

One of the central challenges in IRL is that a given sequence of actions typically can be explained by many different goals. That is, there may be multiple reward functions that would produce the same policy under a given behavioural model. This means that the goals of an agent are only *partially identifiable*. The nature of this partial identifiability in turn depends on the behavioural model. For some behavioural models, the partial identifiability has been studied (Ng & Russell, 2000; Dvijotham & Todorov, 2010; Cao et al., 2021; Kim et al., 2021; Skalse et al., 2022; Schlaginhaufen & Kamgarpour, 2023; Metelli et al., 2023). However, this existing work has focused on a small number of behavioural models that are popular in the current IRL literature. For other plausible behavioural models, the issue of partial identifiability has largely not been studied.

One of the most important parts of a behavioural model is the choice of the *discount function*. In a sequential decision problem, different actions may lead the agent to receive more or less reward at different points in time. In these cases, it is common to let the agent discount future reward, so that reward which will be received sooner is given greater weight than reward which will be received later. There are multiple reasons for doing this. Some reasons are very practical; discounting typically leads to more stable behaviour, and it can be used to ensure that the preferences of an agent are well-defined even over infinite time horizons. Other reasons are more philosophical in nature; a plan

is always generated within some model of the environment, and there is always a risk that this model is wrong. The longer a plan is, the greater is the risk that a mistaken assumption in the model will force the plan to be revised. This uncertainty can be accounted for using discounting. In the context of IRL, there is also an empirical motivation; human preferences are generally well-described as involving discounting of future reward. People are impatient, and all other things being equal, it is better to receive a benefit sooner rather than later ("a bird in the hand is worth two in the bush"). For a more in-depth overview, see e.g. Frederick et al. (2002).

Discounting can be done in many different ways. The two most prominent and widely discussed forms of discounting are *exponential discounting*, according to which reward received at time $t$ is given weight $\gamma^t$, and *hyperbolic discounting*, according to which reward received at time $t$ is given weight $1/(1 + kt)$. Here $\gamma \in (0, 1]$ and $k \in (0, \infty)$ are two parameters. At the moment, most work on IRL assumes that the observed agent discounts exponentially. The main reason for this is that exponential discounting has many convenient theoretical properties. For example, it can be computed recursively, and it leads to preferences that are consistent over time (an issue we will return to later). However, there is extensive work in the behavioural sciences which suggests that humans (and many other animals) are better modelled as using hyperbolic discounting (e.g. Thaler, 1981; Mazur, 1987; Green & Myerson, 1996; Kirby, 1997; Frederick et al., 2002). It is therefore a significant limitation that IRL primarily uses behavioural models with exponential discounting. If humans discount hyperbolically, then these models are *misspecified* (Skalse et al., 2022).

In this paper, we study the issue of partial identifiability in IRL with non-exponential discounting. Specifically, we introduce a number of behavioural models for agents with non-exponential discounting, and characterise the partial identifiability of these models. Moreover, we show that IRL in these cases is unable to infer enough information about $R$ to identify the correct optimal policy. This suggests that IRL alone is insufficient to adequately characterise the preferences of an agent that uses non-exponential discounting. All our results apply to agents that discount hyperbolically, but most of our results also apply to agents that use other types of (non-exponential) discounting.

## 1.1 RELATED WORK

The issue of partial identifiability in IRL has been studied for many behavioural models. In particular, Ng & Russell (2000) study optimal policies with state-dependent reward functions, Dvijotham & Todorov (2010) study regularised MDPs with a particular type of dynamics, Cao et al. (2021) study how the reward ambiguity can be reduced by combining information from multiple environments, Skalse et al. (2022) study three different behavioural models and introduce a framework for reasoning about partial identifiability in reward learning, Schlaginhaufen & Kamgarpour (2023) study ambiguity in constrained MDPs, and Metelli et al. (2023) quantify sample complexities for optimal policies. However, all these papers assume exponential discounting.

Most IRL algorithms are designed for agents that discount exponentially, but some papers have considered hyperbolic discounting (Evans et al., 2015; Chan et al., 2019; Schultheis et al., 2022). However, these papers do not formally characterise the identifiability of $R$ given their algorithms.

## 2 PRELIMINARIES

In this section, we give a brief overview of all material that is required to understand this paper, together with our basic assumptions, and our choice of terminology.

### 2.1 REINFORCEMENT LEARNING

A *Markov Decision Processes* (MDP) is a tuple $(\mathcal{S}, \mathcal{A}, \tau, \mu_0, R, \gamma)$ where $\mathcal{S}$ is a set of *states*, $\mathcal{A}$ is a set of *actions*, $\tau : \mathcal{S} \times \mathcal{A} \to \Delta(\mathcal{S})$ is a *transition function*, $\mu_0 \in \Delta(\mathcal{S})$ is an *initial state distribution*, $R : \mathcal{S} \times \mathcal{A} \times \mathcal{S} \to \mathbb{R}$ is a *reward function*, and $\gamma \in (0, 1]$ is a *discount rate*. In this paper, we assume that $\mathcal{S}$ and $\mathcal{A}$ are finite. A *policy* is a function $\pi : (\mathcal{S} \times \mathcal{A})^\star \times \mathcal{S} \to \Delta(\mathcal{A})$. If a policy $\pi$ can be expressed as a function $\mathcal{S} \to \Delta(\mathcal{A})$, then we say that it is *stationary*. We use $\mathcal{R}$ to denote the set of all reward functions definable over $\mathcal{S}$ and $\mathcal{A}$ (i.e. $\mathbb{R}^{\mathcal{S} \times \mathcal{A} \times \mathcal{S}}$), and $\Pi$ to denote the set of all policies that can be defined over $\mathcal{S}$ and $\mathcal{A}$ (i.e. $\Delta(A)^{(\mathcal{S} \times \mathcal{A})^\star \times \mathcal{S}}$).

A *trajectory* $\xi = \langle s_0, a_0, s_1 \ldots \rangle$ is a possible path in an MDP. The *return function* $G$ gives the cumulative discounted reward of a trajectory, $G(\xi) = \sum_{t=0}^{\infty} \gamma^t R(s_t, a_t, s_{t+1})$, the *value function* $V^\pi : \mathcal{S} \to \mathbb{R}$ of a policy encodes the expected cumulative discounted reward from each state when following that policy, and its $Q$-function $Q^\pi : \mathcal{S} \times \mathcal{A} \to \mathbb{R}$ is $Q^\pi(s, a) = \mathbb{E}_{S' \sim \tau(s,a)} [R(s, a, S') + \gamma V^\pi(S')]$. The *advantage function* $A^\pi$ is $Q^\pi - V^\pi$. The *evaluation function* $\mathcal{J}$ gives the expected trajectory return given a policy, $\mathcal{J}(\pi) = \mathbb{E}_{S_0 \sim \mu_0} [V^\pi(S_0)]$. If a policy $\pi$ satisfies that $V^\pi(s) \geq V^{\pi'}(s)$ for all states $s$ and all policies $\pi'$, then we say that $\pi$ is an *optimal policy*. $Q^\star$ denotes the $Q$-function of the optimal policies. This function is unique, even when there are multiple optimal policies.

A state $s_t$ is *terminal* if $\tau(s_t, a) = s_t$ and $R(s_t, a, s_t) = 0$ for all actions $a$. Moreover, an MDP is *episodic* if has one or more terminal states, and every policy with probability 1 eventually enters a terminal state. When talking about episodic MDPs, we implicitly restrict $\mathcal{R}$ to reward functions such that $R(s_t, a, s_t) = 0$ for all terminal states.

When constructing examples of MDPs, it will sometimes be convenient to let the set of actions $\mathcal{A}$ vary between different states. In these cases, we may assume that each state has a "default action" (chosen from the actions available in that state), and that all action which are unavailable in that state simply are equivalent to the default action. For terminal states, we omit the actions completely.

## 2.2 INVERSE REINFORCEMENT LEARNING

In IRL, we wish to infer a reward function $R$ based on a policy $\pi$ that has been computed from $R$. To do this, we need a *behavioural model* that describes how $\pi$ relates to $R$. One of the most common models is *Boltzmann Rationality* (e.g. Ramachandran & Amir, 2007), given by $\mathbb{P}(\pi(s) = a) \propto e^{\beta Q^\star(s,a)}$, where $\beta$ is a temperature parameter, and $Q^\star$ is the optimal $Q$-function for exponential discounting with some fixed discount parameter $\gamma$. An IRL algorithm infers $R$ from $\pi$ by inverting a behavioural model. There are many algorithms for doing this (e.g. Ng & Russell, 2000; Ramachandran & Amir, 2007; Haarnoja et al., 2017, and many others), but for the purposes of this paper, it will not be important to be familiar with the details of how these algorithms work.

## 2.3 PARTIAL IDENTIFIABILITY

Following Skalse et al. (2022), we will characterise partial identifiability in terms of equivalence relations on $\mathcal{R}$. Let us first introduce a number of definitions:

**Definition 1.** A *behavioural model* is a function $\mathcal{R} \to \Pi$.

For example, we could consider a function $b_{\beta,\tau,\gamma}$ that, given a reward $R$, returns the Boltzmann-rational policy with temperature $\beta$ in the MDP $\langle \mathcal{S}, \mathcal{A}, \tau, \mu_0, R, \gamma \rangle$. Note that we consider the environment dynamics (i.e. the transition function, $\tau$) to be part of the behavioural model. This makes it easier to reason about if and to what extent the identifiability of $R$ depends on $\tau$.

**Definition 2.** Given a behavioural model $f : \mathcal{R} \to \Pi$, we say that the *ambiguity* $\mathrm{Am}(f)$ of $f$ is the partition of $\mathcal{R}$ given by the equivalence relation $\equiv_f$, where $R_1 \equiv_f R_2$ if and only if $f(R_1) = f(R_2)$.

Ambiguity partitions can be used to characterise the partial identifiability of different behavioural models. To see this, let us first build an abstract model of an IRL algorithm. Let $R^\star$ be the true reward function. We model the data source as a function $f : \mathcal{R} \to \Pi$, so that the learning algorithm observes the policy $f(R^\star)$. A reasonable learning algorithm should converge to a reward function $R_H$ that is compatible with the observed policy, i.e. a reward such that $f(R_H) = f(R^\star)$. This means that the invariance partition of $f$ groups together all reward functions that the learning algorithm could converge to.

$\mathrm{Am}(f)$ can, by itself, be abstract and difficult to interpret. Therefore, in order to contextualise the partial identifiability of a given behavioural model, it is useful to relate it to other more familiar equivalence relations on $\mathcal{R}$. As such, we will introduce one more definition:

**Definition 3.** Given a behavioural model $f : \mathcal{R} \to \Pi$ and a partition $P$ of $\mathcal{R}$, we say that $f$ is *$P$-identifiable* if $f(R_1) = f(R_2) \implies R_1 \equiv_P R_2$.

In other words, $f$ is $P$-identifiable if the output of $f$ allows us to identify the $P$-class of the true reward function. To make this more intuitive, suppose $P$ is the partitioning such that $R_1 \equiv_P R_2$

when $R_1$ and $R_2$ have the same optimal policies in some environment. Suppose also that we have some behavioural model $f$. In this case, $f$ being $P$-identifiable means that if $f(R_1) = f(R_2)$, then $R_1$ and $R_2$ have the same optimal policies. Thus, if there is a true reward function $R^\star$, and we learn a reward function $R_H$ such that $f(R^\star) = f(R_H)$, then we can be sure that $R_H$ has the same optimal policies as $R^\star$. Similarly, $P$ could also be replaced with other partitionings. In this way, we can qualitatively characterise the ambiguity of $f$ in terms of $P$-identifiability.

## 3  THE NON-EXPONENTIAL SETTING

In order to study IRL with non-exponential discounting, we must first generalise the basic RL setting. We will allow a *discount function* to be any function $d : \mathbb{N} \to [0, 1]$. Some noteworthy examples of discount functions include *exponential discounting*, where $d(t) = \gamma^t$, *hyperbolic discounting*, where $d(t) = 1/(1 + k \cdot t)$, and *bounded planning*, where $d(t) = 1$ if $t \leq n$, else 0. Here $\gamma$, $k$, and $n$ are parameters. In this paper, we are especially interested in hyperbolic discounting, but most of our results apply to arbitrary discount functions.[1]

Many of the basic definitions in RL can straightforwardly be extended to general discount functions. We consider an MDP to be a tuple $\langle \mathcal{S}, \mathcal{A}, \tau, \mu_0, R, d \rangle$, where $d$ may be any discount function, and we define the trajectory return function as $G(\xi) = \sum_{t=0}^{\infty} d(t) \cdot R(\xi_t)$. We say that $V^\pi(\xi)$ is the expected future discounted reward if you start at trajectory $\xi$ and sample actions from $\pi$, and that $Q^\pi(\xi, a)$ is the expected future discounted reward if you start at trajectory $\xi$, take action $a$, and then sample all subsequent actions from $\pi$. Similarly, $\mathcal{J}(\pi) = \mathbb{E}_{S_0 \sim \mu_0}[V^\pi(S_0)]$. As usual, if $\pi$ is stationary, then we let $V^\pi$ and $Q^\pi$ be parameterised by the current state, instead of the past trajectory.

However, other concepts are less straightforward to extend to the RL setting with general discount functions. To start with, for exponential discounting where $\gamma < 1$, we have that $\sum_{t=0}^{\infty} \gamma^t < \infty$. This ensures that $V^\pi$ always is strictly finite for any choice of $R$ and $\tau$. However, if $\sum_{t=0}^{\infty} d(t)$ diverges, then $V^\pi$ will also diverge for some $R$ and $\tau$, which of course is problematic for policy selection. Therefore, it could be reasonable to impose the requirement that $\sum_{t=0}^{\infty} d(t) < \infty$ as a condition on $d$. Unfortunately, this would rule out the relevant hyperbolic discount function. Since this function is of particular interest, we will instead impose conditions on the transition function $\tau$. In particular, if $\tau$ is *episodic*, then $V^\pi$ is always finite, regardless of which discount function is chosen.

**Proposition 1.** *In any episodic MDP, $|V^\pi(s)| < \infty$ for all policies $\pi$ and all states $s$.*

All proofs can be found in the Appendix. Since episodic environments have convergent policy values for all discount functions, our results will assume that the environment is episodic.

An important property of general discount functions is that they can lead to preferences that are *inconsistent over time*. To understand this, consider the following example:

**Example 1.** *Let* Gym *be the MDP* $\langle \mathcal{S}, \mathcal{A}, \tau, \mu_0, R, \gamma \rangle$ *where* $\mathcal{S} = \{s_0, s_1, s_2, s_3\}$, $\mathcal{A} = \{buy, exercise, enjoy, go\ home\}$, $\mu_0 = s_0$, *and the transition function* $\tau$ *is the deterministic function given by the following labelled graph:*

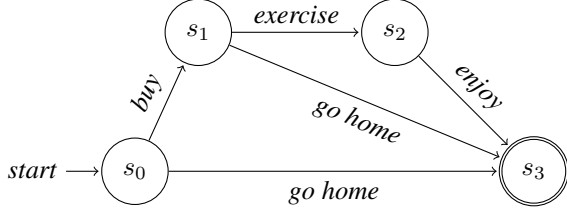

*The discount function* $d$ *is the hyperbolic discount function,* $d(t) = 1/(1 + t)$, *and R is the reward function given by* $R(buy) = -1$, $R(exercise) = -16$, $R(enjoy) = 30$, *and* $R(go\ home) = 0$.

This is a deterministic, episodic environment with four states $\{s_0, s_1, s_2, s_3\}$, where $s_0$ is initial and $s_3$ is terminal. In state $s_0$, the agent can choose between either buying a gym membership, or going

---

[1]Note that average-reward reinforcement learning (Mahadevan, 1996) is not covered by this setting. Averaging the rewards is not a form of discounting, but is instead an alternative to discounting.

home. If it buys the gym membership, then it gets to choose between exercising at the gym, or going home. If it exercises, then it gets to enjoy the benefits of exercise, after which the episode ends. Similarly, if the agent ever goes home, the episode also ends.

We can calculate the value of each trajectory from the initial state $s_0$; $G(\text{go home}) = 0$, $G(\text{buy, go home}) = -1$, and $G(\text{buy, exercise, enjoy}) = 1$. This means that the most valuable trajectory from $s_0$ involves buying a gym membership, and then exercising. However, if we calculate the value of each trajectory from state $s_1$, we (paradoxically) find that $G(\text{go home}) = 0$ and $G(\text{exercise, enjoy}) = -1$. This means that the agent at state $s_0$ would prefer to buy a gym membership, and then exercising. However, after having bought the gym membership, the agent now prefers to go home instead of exercising. In other words, the agent has preferences that are inconsistent over time. We can formalise this as follows.

**Definition 4.** A discount function $d$ is *temporally consistent* if for all sequences $\{x_t\}_{t=0}^{\infty}, \{y_t\}_{t=0}^{\infty}$, $\sum_{t=0}^{\infty} d(t) \cdot x_t < \sum_{t=0}^{\infty} d(t) \cdot y_t$ implies that $\sum_{t=0}^{\infty} d(t+n) \cdot x_t < \sum_{t=0}^{\infty} d(t+n) \cdot y_t$ for all $n$.

Intuitively, if a discount function $d$ is temporally consistent, and at some time $n$, it prefers a sequence of rewards $\{x_t\}_{t=0}^{\infty}$ over another sequence $\{y_t\}_{t=0}^{\infty}$, then this is also true at every other time $n$. On the other hand, if $d$ is *not* temporally consistent, then it may change its preference as time passes. We can see this in the MDP from Example 1.It is easy to show that exponential discounting *is* temporally consistent, and Example 1 demonstrates that hyperbolic discounting is *not* temporally consistent.[2] What about other discount functions? As it turns out, exponential discounting is the *only* form of discounting that is temporally consistent. This means that *all other discount functions* can lead to preferences that are not consistent over time.

**Proposition 2.** $d$ *is temporally consistent if and only if* $d(t) = \alpha\gamma^t$ *for some* $\alpha, \gamma \in [0, 1]$.

For a proof of Proposition 2, see Strotz (1955) or Lattimore & Hutter (2014). This result indirectly implies that there no longer is an unamibiguous notion of what it means for a policy to be "better" than another policy in the setting with non-exponential discounting. For instance, in Example 1, should the best policy choose to exercise at $s_1$, or should it choose to go home? There are multiple ways to answer this question, which in turn means that there are multiple ways to formalise what it means for an agent to "use" hyperbolic discounting (or other non-exponential discount functions). In the next section, we explore several ways of dealing with this issue.

## 4    BEHAVIOURAL MODELS

We wish to construct behavioural models for agents that use non-exponential discounting. To do this, we must first decide what it means for a policy to be "optimal" in this setting. Because of temporal inconsistency, the ordinary notion of optimality does not automatically apply, and there are multiple ways to extend the concept. Accordingly, we introduce three new definitions:

**Definition 5.** A policy $\pi$ is *weakly resolute* if there is no policy $\pi'$ such that $\mathcal{J}(\pi) < \mathcal{J}(\pi')$. It is *strongly resolute* if there is no $\pi'$ or $\xi$ such that $V^{\pi}(\xi) < V^{\pi'}(\xi)$.

A resolute policy maximises expected reward as calculated from the initial state. In other words, it effectively ignores the fact that its preferences are changing over time, and instead always sticks to the preferences that it had at the start. In Example 1, a resolute policy would buy a gym membership, and then exercise. The difference between a strongly resolute policy and weakly resolute policy is analogous to the difference between an optimal policy and a policy that maximises $\mathcal{J}$. A strongly resolute policy is always weakly resolute, but a weakly resolute policy may not be strongly resolute if (for example) it would take a sub-optimal action in a state visited with probability zero.

**Definition 6.** A policy $\pi$ is *naïve* if for each trajectory $\xi$, if $a \in \text{supp}(\pi(\xi))$, then there is a policy $\pi^{\star}$ such that $\pi^{\star}$ maximises $V^{\pi^{\star}}(\xi)$ and $a \in \text{supp}(\pi^{\star}(\xi))$.

A naïve policy ignores the fact that its preferences may not be temporally consistent. Rather, in each state, it computes a policy that is resolute from that state, and then takes an action that this policy

---

[2]Note that this behaviour is an important reason for why hyperbolic discounting is considered to be a good fit for human data. Under experimental conditions, humans can exhibit *preference reversals* in a way that is consistent with hyperbolic discounting (see e.g. Frederick et al., 2002).

would have taken, without taking into account that it may not actually follow this policy later. In Example 1, a naïve policy would buy a gym membership, but then go home without exercising.

**Definition 7.** A policy $\pi$ is *sophisticated* if $\text{supp}(\pi(\xi)) \subseteq \text{argmax}Q^\pi(\xi, a)$ for all trajectories $\xi$.

A sophisticated policy is aware that its preferences are temporally inconsistent, and acts accordingly. Specifically, $\pi$ is sophisticated if it only takes actions that are optimal *given that all subsequent actions are sampled from $\pi$*. In Example 1, a sophisticated policy would choose to not exercise in state $s_1$. In state $s_0$, it realises that it in $s_1$ would choose to go home instead of exercising. Since it in $s_0$ prefers to go home over buying a gym membership and then going home, it chooses to go home without buying a membership.

If $d(t) = \gamma^t$, then Definitions 5-7 are all essentially equivalent to optimality. Formally:

**Theorem 1.** *In an MDP with exponential discounting, the following are equivalent: (1) $\pi$ is optimal, (2) $\pi$ is strongly resolute, (3) $\pi$ is naïve, and (4) $\pi$ is sophisticated. Moreover, (5) $\pi$ is weakly resolute and (6) $\pi$ maximises $\mathcal{J}(\pi)$ are also equivalent, and (1)-(4) imply (5)-(6).*

However, while Definitions 5-7 are all equivalent under exponential discounting, they can be quite different if other forms of discounting are used, as already discussed for Example 1. As such, each of these definitions give us a reasonable way to extend the notion of an "optimal" policy to the setting with general discount functions. In the coming sections, we will discuss each type of policy in turn, and show how to use them to define new behavioural models.

## 4.1 RESOLUTE POLICIES

In this section, we will show how to construct behavioural models based on resolute policies. First of all, we note that any episodic MDP is guaranteed to have a resolute policy, regardless of what discount function it uses. Moreover, there is always at least one resolute policy that is deterministic.

**Theorem 2.** *In any episodic MDP, there exists a deterministic strongly resolute policy.*

However, while an exponentially discounted MDP always has an optimal policy that is stationary, there may not be any stationary resolute policy. Note that this is a consequence of the fact that non-exponential discounting can lead to preferences that are not temporally consistent.

**Proposition 3.** *There are episodic MDPs with no stationary (strongly or weakly) resolute policies.*

To be well-defined, a behavioural model must pick a unique policy $\pi$ for each reward function $R$. However, there are reward functions for which multiple policies are resolute,[3] and so we need a criterion for choosing between them. Fortunately, this can be done in a relatively straightforward way. First, we need to define the "resolute $Q$-function". Note that since resolute policies may have to be non-stationary, this $Q$-function must depend on the current time (in addition to the state):

**Definition 8.** Given an episodic MDP, the *resolute $Q$-function $Q^R : \mathcal{S} \times \mathbb{N} \times \mathcal{A} \to \mathbb{R}$* is defined by letting $Q^R(s, t, a)$ equal $\max_{\pi, \xi} V^\pi(\xi)$ given that $\xi$ has length $t$ and ends in $s$, and that $\pi(\xi) = a$.

**Proposition 4.** *In any episodic MDP, the resolute $Q$-function $Q^R$ exists and is unique.*

$Q^R(s, t, a)$ is the greatest amount of expected cumulative discounted reward (as evaluated from time 0) obtainable from state $s$ at time $t$, conditional on first taking action $a$. Any policy that always takes an action that maximises $Q^R(s, t, a)$ when it visits state $s$ at time $t$, is (strongly) resolute.

Since $Q^R$ is unique, it can be used to define behavioural models for resolute agents. A natural choice would be to always mix uniformly among all actions that maximise $Q^R$, but we could also break ties using some fixed rule. Alternatively, if we want a noisily resolute agent, then we could apply the softmax function to $Q^R$, or we could let the agent take a random action with probability $\epsilon$, etc. To capture all of these options, we introduce the following definition:

**Definition 9.** The *resolute advantage function $A^R : \mathcal{S} \times \mathbb{N} \times \mathcal{A} \to \mathbb{R}$* is given by $A^R(s, t, a) = Q^R(s, t, a) - \max_{a'} Q^R(s, t, a')$. A behavioural model $f : \mathcal{R} \to \Pi$ is *regularly resolute* if $f(R_1) = f(R_2)$ whenever $A_1^R = A_2^R$.

Most natural ways to specify policies for the resolute objective will satisfy Definition 9.

---

[3]For example, every policy is (strongly) resolute for the reward function that is 0 everywhere.

## 4.2 NAÏVE POLICIES

Here, we show how to construct behavioural models based on naïve policies. We begin by noting that any episodic MDP is guaranteed to have a naïve policy that is both deterministic and stationary:

**Theorem 3.** *In any episodic MDP, there exists a stationary, deterministic, naïve policy.*

A behavioural model must pick a unique policy for each reward function, and there can be multiple naïve policies.[4] Therefore, we need a criterion for choosing between them. As for resolute policies, this can fortunately be done in a straightforward way. We first define the "naïve $Q$-function":

**Definition 10.** The *naïve Q-function* $Q^{\mathrm{N}} : \mathcal{S} \times \mathcal{A} \to \mathbb{R}$ is defined as $Q^{\mathrm{N}}(s, a) = Q^{\mathrm{R}}(s, 0, a)$.

**Proposition 5.** *In any episodic MDP, the naïve Q-function $Q^{\mathrm{N}}$ exists and is unique.*

$Q^{\mathrm{N}}(s, a)$ is the greatest amount of reward obtainable from state $s$, conditional on first taking action $a$. Any policy which in each state $s$ takes an action that maximises $Q^{\mathrm{N}}(s, a)$ is naïve.

Since $Q^{\mathrm{N}}$ is unique, it can be used to define behavioural models for resolute agents. For example, we can mix uniformly among all actions that maximise $Q^{\mathrm{N}}$, or we could apply the softmax function to $Q^{\mathrm{N}}$, etc. To capture all these options, we introduce the following definition:

**Definition 11.** The *naïve advantage function* $A^{\mathrm{N}} : \mathcal{S} \times \mathcal{A} \to \mathbb{R}$ is given by $A^{\mathrm{N}}(s, a) = Q^{\mathrm{N}}(s, a) - \max_{a'} Q^{\mathrm{N}}(s, a')$. A behavioural model $f : \mathcal{R} \to \Pi$ is *regularly naïve* if $f(R_1) = f(R_2)$ whenever $A_1^{\mathrm{N}} = A_2^{\mathrm{N}}$.

Most natural ways to specify policies for the naïve objective will satisfy Definition 11.

## 4.3 SOPHISTICATED POLICIES

In this section, we will show how to construct behavioural models based on sophisticated policies. We begin by noting that any episodic MDP always has a stationary sophisticated policy:

**Theorem 4.** *In any episodic MDP, there exists a stationary sophisticated policy.*

However, while there is always a stationary sophisticated policy, there are environments where all sophisticated policies are nondeterministic. Intuitively, this is again a consequence of temporal inconsistency (if an agent wishes to do one thing on its first visit to a state $s$, and a different thing on subsequent visits to $s$, then this may compel it to randomise its actions at $s$).

**Proposition 6.** *There exists episodic MDPs in which every sophisticated policy is nondeterministic.*

A behavioural model must pick a unique policy for each reward function, and there can be multiple sophisticated policies.[5] Therefore, we need a criterion for choosing between them. Unfortunately, this is less straightforward for sophisticated policies than it is for resolute and naïve policies. The reason for this is that there is no unique "sophisticated $Q$-function":

**Proposition 7.** *There exists episodic MDPs $M$ with policies $\pi_1, \pi_2$ such that both $\pi_1$ and $\pi_2$ are sophisticated in $M$, but $Q^{\pi_1} \neq Q^{\pi_2}$.*

As such, there are MDPs with multiple sophisticated policies, and where none of them can reasonably be said to be "more canonical" than the others. This makes it more difficult to say how sophisticated agents should pick their policies. Intuitively speaking, we want it to be the case that if $R_1$ and $R_2$ are sufficiently similar, then $f(R_1) = f(R_2)$. We can capture this intuitive requirement with the following condition:

**Definition 12.** A behavioural model $f : \mathcal{R} \to \Pi$ is *regularly sophisticated* if there for each reward $R_1$ exists a policy $\pi_1$ such that if $R_2$ is a reward with $A_2^{\pi_1} = A_1^{\pi_1}$, then $f(R_1) = f(R_2)$.

Let us briefly unpack this definition. Here $\pi_1$ is intended to be a policy that is sophisticated under $R_1$ (but our later results will not rely on the assumption that this is the case). As such, Definition 12 roughly says that if $f$ prefers $\pi_1$ for $R_1$, and $\pi_1$ has the same advantage function under $R_2$ as it does for $R_1$, then $f$ should also prefer $\pi_1$ for $R_2$. Also note that Definition 12 permits, but does not require, that $f(R_1) = \pi_1$. This means that Definition 12 can capture behavioural models that select noisy policies (by applying the softmax function to $A^{\pi}$, for example).

---

[4]For example, every policy is naïve for the reward function that is 0 everywhere.

[5]For example, every policy is sophisticated for the reward function that is 0 everywhere.

## 5 IDENTIFIABILITY

We have shown how to define behavioural models for inverse reinforcement learning with non-exponential discounting. In this section, we analyse whether or not these models are identifiable. To do this, we must first select an appropriate equivalence relation on $\mathcal{R}$. We have chosen to carry out our analysis in terms of the equivalence relation $\equiv_{\mathrm{OPT}^{\tau,\gamma}}$, according to which $R_1 \equiv_{\mathrm{OPT}^{\tau,\gamma}} R_2$ if the (exponentially discounted) MDPs $(\mathcal{S}, \mathcal{A}, \tau, \mu_0, R_1, \gamma)$ and $(\mathcal{S}, \mathcal{A}, \tau, \mu_0, R_2, \gamma)$ have the same *optimal policies*.[6] We assume that $\equiv_{\mathrm{OPT}^{\tau,\gamma}}$ and the behavioural model $f_\tau$ is defined in terms of the same transition function $\tau$, and that $\gamma \in (0, 1]$. In other words, this means that we will consider a behavioural model to be indentifiable if the policy that this model computes in a given environment is always sufficient to determine what policy would be optimal in that same environment.

This choice may not be obvious – if we observe a policy that is computed with non-exponential discounting, why should we assume that the goal is to compute a policy that is optimal under exponential discounting? As we see it, there are several considerations that make this a natural choice. First of all, as we have noted, exponential discounting is the only discount function that always leads to temporally consistent preferences (see Proposition 2). This makes the exponential discount function a canonical choice. Secondly, since we assume that the environment is episodic, we can allow $\gamma = 1$. This means that the undiscounted case is included as a special case of exponential discounting, which is also a canonical choice.

Moreover, as we have noted before, we are especially interested in the hyperbolic discount function. This discount function, in turn, limits to exponential discounting. Specifically, the further away in time a choice is, the more "patient" a hyperbolically discounting agent becomes, until it eventually starts to behave like an undiscounting agent. Moreover, this is also true for an exponentially discounting agent, as $\gamma \to 1$. To state this formally, let $\mathcal{J}_d$ be the policy evaluation function that uses discount function $d$. We then have:

**Theorem 5.** *Assume we have an episodic MDP, let $u(t) = 1$, and let $\pi_1$ and $\pi_2$ be policies such that $\mathcal{J}_u(\pi_1) > \mathcal{J}_u(\pi_2)$. Then if $h(t) = 1/(1 + k \cdot t)$, then there exist an $N \in \mathbb{N}$ such that for all $n \geq N$, if $h^{+n}(t) = h(t+n)$, we have $\mathcal{J}_{h^{+n}}(\pi_1) > \mathcal{J}_{h^{+n}}(\pi_2)$. Moreover, there is a $\gamma \in (0, 1)$ such that, for all $\gamma' \in [\gamma, 1)$, if $e(t) = \gamma'^t$, then we have that $\mathcal{J}_e(\pi_1) > \mathcal{J}_e(\pi_2)$.*

Note that $\mathcal{J}_u(\pi)$ is the undiscounted value of $\pi$. Theorem 5 thus tells us that if the undiscounted value of $\pi_1$ is higher than the undiscounted value of $\pi_2$, then a hyperbolically discounting agent will eventually prefer $\pi_1$ over $\pi_2$. Moreover, this is also true of an exponentially discounting agent, if $\gamma$ is sufficiently close to 1. This can be interpreted as saying that a hyperbolically discounting agent wants to eventually discount exponentially, but that it may give in to temptations in the short term.

We first show that no regularly resolute, regularly naïve, or regularly sophisticated behavioural model is $\mathrm{OPT}_{\tau,\gamma}$-identifiable for all $\tau$, unless $d$ is equivalent to exponential discounting:

**Theorem 6.** *Let $d$ be a discount function, and let $f_{\tau,d}$ be a behavioural model that is regularly resolute, regularly naïve, or regularly sophisticated, for transition function $\tau$ and discount $d$. Then for any $\gamma \in (0, 1]$, unless there is an $\alpha \in (0, 1]$ such that $d(t) = \alpha\gamma^t$ for all $t \leq |\mathcal{S}| - 2$, there exists a transition function $\tau$ such that $f_\tau$ is not $\mathrm{OPT}_{\tau,\gamma}$-identifiable.*

This theorem is saying that, under mild assumptions, there is an environment with reward functions $R_1$ and $R_2$ such that $f_{\tau,d}(R_1) = f_{\tau,d}(R_2)$, but $R_1$ and $R_2$ have different optimal policies. Therefore, none of the behavioural models we have defined give us enough information about the reward function to determine the optimal policy.

Note that Theorem 6 says that there exists *some* transition function $\tau$ under which $f_{\tau,d}$ is not $\mathrm{OPT}_{\tau,\gamma}$-identifiable. This does, by itself, not rule out the possibility that it might be $\mathrm{OPT}_{\tau,\gamma}$-identifiable for most typical transition functions. Therefore, our next result demonstrates that $f_{\tau,d}$ can be shown to not be $\mathrm{OPT}_{\tau,\gamma}$-identifiable under very mild assumptions on $\tau$, given that $f_{\tau,d}$ is regularly resolute, naïve, or regularly sophisticated. We say that a state $s'$ is *controllable* if there is a non-terminal state $s$ and actions $a_1, a_2$ such that $\mathbb{P}(\tau(s, a_1) = s') \neq \mathbb{P}(\tau(s, a_2) = s')$, and that $\tau$ is *non-trivial* if it has at least one controllable state. Moreover, we say that $\tau$ is *acyclic* if no path that is possible under $\tau$ contains a cycle.

---

[6]Note that the set of optimal policies does not depend on the initial state distribution, and so $\equiv_{\mathrm{OPT}^{\tau,\gamma}}$ does not need to be parameterised by $\mu_0$.

**Theorem 7.** *Let $d$ be a discount function, let $\tau$ be a non-trivial acyclic transition function, and let $f_{\tau,d}$ be a behavioural model that is regularly resolute, regularly naïve, or regularly sophisticated, for transition function $\tau$ and discount $d$. Then for any $\gamma \in (0,1]$, unless $\gamma = d(1)/d(0)$, we have that $f_{\tau,d}$ is not $\mathrm{OPT}_{\tau,\gamma}$-identifiable.*

Nearly all transition functions are non-trivial, so Theorem 7 applies very broadly. Note that Theorem 6 makes weaker assumptions about the discount function but stronger assumptions about the transition function, whereas Theorem 7 makes stronger assumptions about the discount function but weaker assumptions about the transition function. In the appendix, we also discuss the question of how to strengthen Theorem 6 and 7.

## 6    DISCUSSION AND FURTHER WORK

In this paper, we analyse partial identifiability in IRL with non-exponential discounting, including (but not limited to) hyperbolic discounting. To this end, we have introduced three types of policies (resolute policies, naïve policies, and sophisticated policies) that generalise the standard notion of optimality to non-exponential discount functions, and shown that these policies always exist in any episodic MDP. We have then analysed the identifiability of these models, and found that each of them, for any discount function, can be too ambiguous to identify the correct optimal policy.

Our results suggest that the preferences of a hyperbolically-discounting agent cannot be completely inferred from their actions alone. IRL always leaves a certain irreducible ambiguity, that persists even in the limit of infinite information – this is true regardless of what type of policy the observed agent uses. However, if the observed agent is optimal or Boltzmann Rational, then this ambiguity is not fundamentally problematic, since all reward functions in the feasible set produce the same optimal policies (e.g. Skalse et al., 2022). By contrast, our results suggest that the ambiguity for agents that use non-exponential discounting is more problematic, in that we may be unable to identify the correct optimal policy even in the limit of infinite information. This imposes certain limitations on the scope of IRL as a tool for preference elicitation.

There are a several ways that our work can be extended to. Improving our understanding of identifiability in IRL is of crucial importance, if we want to use IRL (and similar techniques) as a tool for preference elicitation. Moreover, this analysis must consider behavioural models that are actually realistic: here we have considered hyperbolic discounting, since this is widespread in the behavioural sciences, but there are also many other ways to make our models more plausible. For example, it would be interesting to incorporate models of human risk-aversion, such as prospect theory (Kahneman & Tversky, 1979).

To be informative, the ambiguity of a behavioural model must generally be related to some relevant property of the reward function, i.e., an equivalence relation on $\mathcal{R}$. Under this assumption, we have chosen to examine whether or not our models are too ambiguous to identify the optimal policy of the true reward function. However, there might be other equivalence relations that would also be informative: hence, finding other appropriate equivalence relations by which to judge the ambiguity of behavioural models would thus be another interesting direction for further work.

Finally, there are a few ways that our technical results could be made more complete. In particular, it would be interesting to strengthen Theorem 7 by removing the acyclicity assumption – this is topic of ongoing work.

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
