# A PROOFS

In this appendix, we prove all of our theoretical results.

## A.1 CORE LEMMAS

Let $d : \Pi \times \Pi \to \mathbb{R}$ be the function given by $d(\pi_1, \pi_2) = \frac{1}{e^t}$, where $t$ is the length of the shortest trajectory $\xi$ such that $\pi_1(\xi) \neq \pi_2(\xi)$, or 0 if $\pi_1 = \pi_2$.

**Lemma 1.** $(\Pi, d)$ *is a compact metric space.*

*Proof.* We must first show that $d$ is a metric, which requires showing that it satisfies the following:

1. Identity: $d(\pi_1, \pi_2) = 0$ if and only if $\pi_1 = \pi_2$.

2. Positivity: $d(\pi_1, \pi_2) \geq 0$.

3. Symmetry: $d(\pi_1, \pi_2) = d(\pi_2, \pi_1)$.

4. Triangle Inequality: $d(\pi_1, \pi_3) \leq d(\pi_1, \pi_2) + d(\pi_2, \pi_3)$.

It is straightforward to see that 1-3 hold. For 4, let $t$ be the length of the shortest history $h$ such that $\pi_1(h) \neq \pi_3(h)$. Note that if $d(\pi_1, \pi_3) > d(\pi_1, \pi_2)$ and $d(\pi_1, \pi_3) > d(\pi_2, \pi_3)$, then it must be the case that $\pi_1(h) = \pi_2(h)$ for all $h$ of length $\leq t$, and that $\pi_1(h) = \pi_2(h)$ for all $h$ of length $\leq t$. However, this is a contradiction, since it would imply that $\pi_1(h) = \pi_3(h)$ for all $h$ of length $\leq t$. Thus either $d(\pi_1, \pi_3) \leq d(\pi_1, \pi_2)$ or $d(\pi_1, \pi_3) \leq d(\pi_2, \pi_3)$, which in turn implies that $d(\pi_1, \pi_3) \leq d(\pi_1, \pi_2) + d(\pi_2, \pi_3)$.

Thus $d$ is a metric, which means that $(\Pi, d)$ is a metric space. Next, we will prove that $(\Pi, d)$ is compact, using the Heine-Borel theorem. To do this, we must show that $(\Pi, d)$ is totally bounded and complete.

To see that $(\Pi, d)$ is totally bounded, let $\epsilon$ be an arbitrary positive real number, and let $t = \ln(1/\epsilon)$, so that $\epsilon = 1/e^t$. Moreover, let $\hat{\Pi}$ be the set of all policies that always take action $a_1$ after time $t$ (but which may behave arbitrarily before time $t$). Now $\hat{\Pi}$ is finite, and for every policy $\pi_1$ there is a policy $\pi_2 \in \hat{\Pi}$ such that $d(\pi_1, \pi_2) \leq \epsilon$ (given by letting $\pi_2(\xi) = \pi_1(\xi)$ for all trajectories $\xi$ with length at most $t$). Thus, for every $\epsilon > 0$, $(\Pi, d)$ has a finite cover. Thus $(\Pi, d)$ is totally bounded.

To see that $(\Pi, d)$ is complete, let $\{\pi_i\}_{i=0}^{\infty}$ be a Cauchy sequence. This implies that for every $\epsilon > 0$ there is a positive integer $N$ such that for all $n, m \geq N$ we have $d(\pi_n, \pi_m) < \epsilon$. In our case, this means that there, for each time $t$ is a positive integer $N$ such that for all $n, m \geq N$, we have that $\pi_n(\xi) = \pi_m(\xi)$ for all trajectories $\xi$ shorter than $t$ steps. We can thus define a policy $\pi_\infty$ by letting $\pi_\infty(\xi) = \delta$ (where $\delta \in \Delta(\mathcal{A})$) if there is an $N$ such that, for all $n \geq N$, we have that $\pi_n(\xi) = \delta$. Now $\lim_{i \to \infty} \{\pi_i\}_{i=0}^{\infty} = \pi_\infty$, and $\pi_\infty \in (\Pi, d)$. Thus every Cauchy sequence in $(\Pi, d)$ has a limit that is also in $(\Pi, d)$, and so $(\Pi, d)$ is complete.

Thus, by the Heine-Borel theorem, we have that $(\Pi, d)$ is a compact metric space. $\square$

**Lemma 2.** $\langle \mathcal{S}, \mathcal{A}, \tau, \mu_0, R, d \rangle$ *is episodic if and only if there exists $n \in \mathbb{N}, p \in (0, 1]$ such that for any policy $\pi$ and state $s$, if $\pi$ is run from $s$, then after $n$ steps, it will have entered a terminal state with probability at least $p$.*

*Proof.* For the first direction, assume that there exists $n \in \mathbb{N}, p \in (0, 1]$ such that for any policy $\pi$ and any state $s$, if $\pi$ is run from $s$, then after $n$ steps, it will have entered a terminal state with probability at least $p$. Then for any policy $\pi$, we have that $\pi$ after $kn$ steps will have entered a terminal state with probability at least $1 - p^k$. We of course have that $\lim_{k \to \infty} 1 - p^k = 1$, and so $\pi$ will almost surely eventually enter a terminal state. Since $\pi$ was chosen arbitrarily, this means that $\langle \mathcal{S}, \mathcal{A}, \tau, \mu_0, R, d \rangle$ must be terminal.

For the other direction, assume that $\langle \mathcal{S}, \mathcal{A}, \tau, \mu_0, R, d \rangle$ is episodic. Let $\pi$ and $s$ be selected arbitrarily. Since every policy eventually enters a terminal state with probability 1, there must be a trajectory

$s, a_0, s_1, \ldots$ starting in $s$ and ending in a terminal state, such that each transition has positive probability under $\pi$ and $\tau$. Moreover, the *shortest* such trajectory can contain no more than $|\mathcal{S}|$ states – otherwise there must be a loop that occurs with probability 1 (in which case the MDP would not be episodic). Since $\pi$ and $s$ were selected arbitrarily, this shows that there is an $n = |\mathcal{S}| \in \mathbb{N}$ such that for any policy $\pi$ and state $s$, if $\pi$ is run from $s$, then after $n$ steps, it will have entered a terminal state with positive probability. It remains to be shown that this probability is bounded below by some positive constant $p$.

Let $q(\pi, s)$ be the probability that $\pi$ will have entered a terminal state after $n$ steps, starting in state $s$. Note that this function is continuous, when viewed as a function from $(\Pi, d)$ to $[0, 1]$. In particular, if $\pi_1(\xi) = \pi_2(\xi)$ for all trajectories $\xi$ of length at most $n$, then $q(\pi_1, s) = q(\pi_2, s)$. Thus, for every $\epsilon > 0$ there is a $\delta = \ln(1/n)$ such that if $d(\pi_1, \pi_2) < \delta$, then $|q(\pi_1, s) - q(\pi_2, s)| = 0 < \epsilon$. Moreover, by Lemma 1, we have that $(\Pi, d)$ is a compact metric space. Thus, by the extreme value theorem, for each $s$ there is a policy $\pi_s \in \Pi$ that minimises $q(\pi, s)$. Moreover, we have already established that for any policy $\pi$ and state $s$, if $\pi$ is run from $s$, then after $n$ steps, it will have entered a terminal state with positive probability. Thus $q(\pi_s, s) > 0$. Since $\mathcal{S}$ is finite, we can now set $p$ to $\min_s(\pi_s, s)$, and thus complete the proof. $\qquad\square$

### A.2 Convergent Policy Values

In this section, we provide the proofs of the claims regarding convergent policy values.

**Proposition 1.** *If $\langle \mathcal{S}, \mathcal{A}, \tau, \mu_0, R, d \rangle$ is episodic, then we have that $|V^\pi(s)| < \infty$ for all policies $\pi$ and all states $s$.*

*Proof.* As per Lemma 2, in any episodic MDP, there is an $n$ and a $p$ such that for any state $s$ and policy $\pi$, we have that $\pi$ after $n$ steps will have entered a terminal state with probability at least $p$. Moreover, since $\mathcal{S}$ and $\mathcal{A}$ are finite, we have that $m = \max_{s,a,s'} |R(s, a, s')| \leq \infty$. Since $d(t) \in [0, 1]$, this means the discounted reward obtained over any sequence of $n$ steps is at least $-mn$, and at most $mn$. Since the probability of entering a terminal state along any such sequence is at least $p$, we have that

$$|V^\pi(s)| \leq \left( \frac{mn}{1 - p} \right),$$

which is finite. $\qquad\square$

**Proposition 2.** *If $\langle \mathcal{S}, \mathcal{A}, \tau, \mu_0, R_1, d \rangle$ is not episodic, and $\sum_{t=0}^{\infty} d(t) = \infty$, then there is a reward function $R_2$, policy $\pi$, and state $s$, such that $V^\pi(s) = \infty$ in $\langle \mathcal{S}, \mathcal{A}, \tau, \mu_0, R_2, d \rangle$.*

*Proof.* Let $R_2$ be the reward function such that $R_2(s, a, s') = 1$ unless $s$ or $s'$ is terminal. Now, since $\langle \mathcal{S}, \mathcal{A}, \tau, \mu_0, R_1, d \rangle$ is not episodic, there is a policy $\pi$ that, with positive probability, never enters a terminal state. Let this probability be $p$. This means that there must be an initial state $s_0$ such that the probability that $\pi$ never enters a terminal state, conditional on the first state being $s_0$, is at least $p$. This means that $V^\pi(s_0) \geq p \cdot \sum_{t=0}^{\infty} 1 = \infty$ in the MDP $\langle \mathcal{S}, \mathcal{A}, \tau, \mu_0, R_2, d \rangle$. $\qquad\square$

### A.3 Temporal Consistency

**Proposition 3.** *A discount function $d$ is temporally consistent if and only if $d(t) = \alpha\gamma^t$ for some $\alpha, \gamma \in [0, 1]$.*

The proof of this proposition is given in Lattimore & Hutter (2014) (their Theorem 13). Their terminology is slightly different from ours, but their proof applies to our case with essentially no modification.

### A.4 Correspondence To Optimality

Here, we will establish the relationship between optimal policies, resolute policies, naïve policies, and sophisticated policies, in the case of exponential discounting.

**Theorem 1.** *If $\langle \mathcal{S}, \mathcal{A}, \tau, \mu_0, R, \gamma \rangle$ is an MDP with exponential discounting, then the following are equivalent:*

1. $\pi$ is optimal.

2. $\pi$ is strongly resolute.

3. $\pi$ is naïve.

4. $\pi$ is sophisticated.

*Additionally, the following are also equivalent:*

5. $\pi$ is weakly resolute.

6. $\pi$ maximises $\mathcal{J}(\pi)$.

*Moreover, 1-4 imply 5-6.*

*Proof.* First of all, in an exponentially discounted MDP, $\pi_1$ is optimal if for all states $s$ and policies $\pi_2$, we have $V^{\pi_1}(s) \geq V^{\pi_2}(s)$, and $\pi_1$ is strongly resolute if for all states $s$, times $t$, and policies $\pi_2$, we have $V^{\pi_1}(s,t) \geq V^{\pi_2}(s,t)$. Moreover, since exponential discounting is temporally consistent, we have that for all $t$, $V^{\pi_1}(s) \geq V^{\pi_2}(s)$ if and only if $V^{\pi_1}(s,t) \geq V^{\pi_2}(s,t)$. From this it follows that 1 and 2 are equivalent in an exponentially discounted MDP.

Secondly, in an exponentially discounted MDP, we have that a policy $\pi$ is optimal if and only if $\mathrm{supp}(\pi(s)) \subseteq \mathrm{argmax}_a(Q^\star(s,a))$, and $\pi$ is naïve if and only if for each state $s$, if $a \in \mathrm{supp}(\pi(s))$, then there is a policy $\pi^\star$ such that $\pi^\star$ maximises $V^{\pi^\star}(s)$ and $a \in \mathrm{supp}(\pi^\star(s))$. Moreover, if $\pi^\star$ maximises $V^{\pi^\star}(s)$, then each $a \in \mathrm{supp}(\pi^\star(s))$ must maximise $Q^\star$. From this, it follows that 1 and 3 are equivalent in exponentially discounted MDPs.

Furthermore, in an exponentially discounted MDP, we have that a policy $\pi$ is optimal if and only if it is a fixed point under *policy iteration*, and $\pi$ is sophisticated if and only if $\mathrm{supp}(\pi(s)) \subseteq \mathrm{argmax}Q^\pi(s,a)$. From this, it follows that 1 and 4 are equivalent in exponentially discounted MDPs.

Next, note that in an exponentially discounted MDP, 5 and 6 are definitionally directly equivalent. Finally, from the fact that optimal policies are optimal from all initial states, we have that 1-4 imply 5-6. This completes the proof. □

### A.5 RESOLUTE POLICIES

We here provide our proofs about resolute policies.

**Lemma 3.** *In any episodic MDP $\langle \mathcal{S}, \mathcal{A}, \tau, \mu_0, R, d \rangle$, each state $s$ and time $t$, there exists a policy $\pi_1$ such that $V^{\pi_1}(s,t) \geq V^{\pi_2}(s,t)$ for all $\pi_2$.*

*Proof.* We will show that $V^\pi(s,t)$ is continuous, when viewed as a function from $(\Pi, d)$ to $\mathbb{R}$. Let $\pi_1$ be any policy, and $\epsilon$ any positive real number. Since $\mathcal{S}$ and $\mathcal{A}$ are finite, we have $m = \max_{s,a,s'} |R(s,a,s')| < \infty$. Moreover, as per Lemma 2, since the MDP is episodic, there is an $n$ and $p$ such that any policy $\pi$ after $n$ steps will have entered a terminal state with probability at least $p$. Thus, if $\pi_1(\xi) = \pi_2(\xi)$ for all trajectories of length $kn$, then the difference in reward between $\pi_1$ and $\pi_2$ can be at most $mnp^k/(1-p)$. For any $k$ that is sufficiently large (and hence for any $d(\pi_1, \pi_2)$ that is sufficiently small), we have that this quantity is below $\epsilon$. Thus, for every $\epsilon$ there is a $\delta$ such that, if $d(\pi_1, \pi_2) < \delta$ then $|V^{\pi_1}(s,t) - V^{\pi_1}| < \epsilon$. This means that $V^\pi(s,t)$ is continuous, when viewed as a function from $(\Pi, d)$ to $\mathbb{R}$.

By Lemma 1, we have that $(\Pi, d)$ is compact. Thus, by the extreme value theorem, there must exist a policy $\pi_1$ such that $V^{\pi_1}(s,t) \geq V^{\pi_2}(s,t)$ for all $\pi_2$. □

**Proposition 4.** *In any episodic MDP, the resolute $Q$- function $Q^{\mathrm{R}}$ exists and is unique.*

*Proof.* Immediate from Lemma 3. □

**Theorem 2.** *In any episodic MDP, there exists a deterministic strongly resolute policy.*

*Proof.* By Proposition 4, in any episodic MDP, the resolute $Q$- function $Q^{\mathrm{R}}$ exists and is unique. We now have that any policy $\pi$ is strongly resolute if, for each trajectory $\xi$, we have that $\pi(\xi) \in \operatorname{argmax}_a Q^{\mathrm{R}}(s, |\xi|, a)$, where $s$ is the last state in $\xi$. There always exists a deterministic policy satisfying this criterion. $\qquad\square$

**Example 2.** *Let* Loop *be the 4-state MDP where* $\mathcal{S} = \{s_0, s_1, s_2, s_t\}$, $\mathcal{A} = \{up, down\}$, *and* $\mu_0 = s_0$. *We have that* $\tau(s_0, up) = s_1$ *and* $\tau(s_0, down) = s_2$. *For* $s \in \{s_1, s_2\}$, *we have that* $\tau(s, a) = s_0$ *with probability* $0.95$, *and* $s_t$ *with probability* $0.05$, *for both actions* $a \in \mathcal{A}$. *The reward function $R$ is zero everywhere, except that* $R(s_0, up, s_1) = 3$ *and* $R(s_2, a, s') = 5$ *for both* $a \in \mathcal{A}$ *and both* $s' \in \{s_0, s_t\}$. *The discount $d$ is the hyperbolic discount function,* $d(t) = 1/(1+t)$. *This environment can be depicted as:*

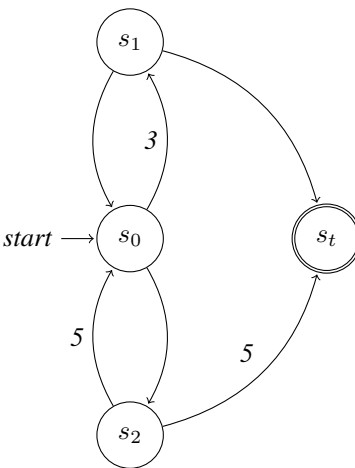

*This MDP repeatedly gives the agent a choice between receiving $2$ reward instantaneously, or $5$ reward in one step, where there is a $5\%$ chance that the episode will end after each choice is made. With hyperbolic discounting, we have that $3d(t) > 5d(t+1)$ if $t = 0$, and that $3d(t) < 5d(t+1)$ for all $t \geq 1$. In other words, the agent would want to pick $3$ reward the first time, and $5$ reward afterwards.*

**Proposition 5.** *There exists episodic MDPs in which every (strongly or weakly) resolute policy is non-stationary.*

*Proof.* Consider the MDP Loop, given in Example 2. We will show that there is a non-stationary policy that outperforms every stationary policy in this MDP, and hence prove that all resolute policies must be non-stationary.

In this MDP, the only state where the agent can make a meaningful choice is in state $s_0$. Assume that $\pi_p$ is the stationary policy that chooses left with probability $p$, and otherwise chooses right. Then $\mathcal{J}(\pi_p)$ is

$$\sum_{i=1}^{\infty} (0.95^i) * (3p/(1+2i) + 5 * (1-p)/(2+2i)).$$

This sum can in turn be equivalently expressed as

$$\frac{1}{38}\Big( -100\log(20)p + 12\sqrt{95}\tanh^{-1}\Big(0.5\sqrt{19/5}\Big)p$$
$$- 19p + 100\log(20) - 95\Big).$$

This expression is maximised on $p \in [0, 1]$ for $p = 0$, in which case $\mathcal{J}(\pi_p) \approx 5.38$. This is thus the highest value obtainable by any stationary policy.

Consider now the policy $\pi$ where $\pi(\xi) = $ left if $|\xi| = 1$, and otherwise returns right (that is, $\pi$ selects left on its first visit to $s_0$, and afterwards selects right). Now $\mathcal{J}(\pi)$ is

$$3 + 0.95 * 5 * \sum_{i=2}^{\infty} (0.95^i/(2+2i)) \approx 6.99.$$

We have thus shown that there is a non-stationary policy $\pi$ such that $\mathcal{J}(\pi) > \mathcal{J}(\pi_p)$ for all stationary policies $\pi_p$. This, in turn, means that all (strongly or weakly) resolute policies in `Loop` must be non-stationary. □

## A.6 Naïve Policies

We here provide our proofs about naïve policies.

**Proposition 6.** *In any episodic MDP, the naïve Q-function $Q^{\mathrm{N}}$ exists and is unique.*

*Proof.* Immediate from Proposition 4. □

**Theorem 3.** *In any episodic MDP, there exists a stationary deterministic naïve policy.*

*Proof.* By Proposition 4, in any episodic MDP, the naïve $Q$- function $Q^{\mathrm{N}}$ exists and is unique. We now have that any policy $\pi$ is naïve if, for each trajectory $\xi$, we have that $\pi(\xi) \in \mathrm{argmax}_a Q^{\mathrm{n}}(s,a)$, where $s$ is the last state in $\xi$. There always exists a stationary deterministic policy satisfying this criterion. □

## A.7 Sophisticated Policies

We here provide our proofs about sophisticated policies.

**Theorem 4.** *In any episodic MDP, there exists a stationary sophisticated policy.*

*Proof.* By the Kakutani fixed-point theorem, if $X$ is a non-empty, convex, and compact subset of a Euclidean space $\mathbb{R}^n$, and $\phi : X \to \mathcal{P}(X)$ is a set valued function with the property that

1. $\phi(x)$ is non-empty, closed, and convex for all $x \in X$, and

2. $\phi$ is upper hemicontinuous,

then $\phi$ has a fixed point.

Let $\hat{\Pi}$ be the set of all stationary policies. We say that a policy $\pi_2$ is a *local improvement* of $\pi_1$ in $s$ if $\mathrm{supp}(\pi_2(s)) \subseteq \mathrm{argmax}_a Q^{\pi_1}(s,a)$. Let $\phi : \hat{\Pi} \to \mathcal{P}(\hat{\Pi})$ be the function that, given $\pi$, returns the set of all policies which are local improvements of $\pi$ in all $s$.

We can begin by noting that $\hat{\Pi}$ of course is a non-empty, convex, and compact subset of the Euclidean space $\mathbb{R}^{|S||A|}$. It is immediate from the definition that $\phi$ is both convex and closed. Moreover, since the MDP is episodic, we have that $Q^\pi(s,a)$ exists (i.e. is finite) for all $\pi, s, a$, by Proposition 1. Since there is a finite number of actions, we thus also have that $\phi(\pi)$ is non-empty.

Claude Berge's Maximum Theorem says that if $X$ and $Y$ are topological spaces, and $f : X \times Y \to \mathbb{R}$ is continuous, and if moreover

1. $f^\star(y) = \sup\{f(x,y) : x \in X\}$

2. $C(y) = \{x : f(x,y) = f^\star(x)\}$

then $f^\star$ is continuous, and $C$ is upper hemicontinuous. Let $X$ and $Y$ both be equal to $\Pi$, and let $f : \Pi \times \Pi \to \mathbb{R}$ be the function where $f(\pi_1, \pi_2) = \sum_s \mathbb{E}_{a \sim \pi_2(s)}[Q^{\pi_1}(s,a)]$. Now $f$ is continuous, and $C(\pi_1) = \{\pi_2 : f(\pi_1, \pi_2) = f^\star(\pi_1)\} = \phi(\pi_1)$. Claude Berge's Maximum Theorem then implies that $\phi$ is upper hemicontinuous.

The Kakutani fixed-point theorem then implies that $\phi$ must have a fixed point, which means that there must be a sophisticated policy. Moreover, by construction, this policy is stationary. □

**Example 3.** *Let* Tempt *be the MDP where $\mathcal{S}$ has 32 states $\{s_0, s_1, \ldots s_{31}\}$, $\mathcal{A} = \{up, down\}$, and $\mu_0 = s_0$. For $i \in 2 \ldots 30$, we have that $\tau(s_i, a) = s_{i+1}$ for both $a \in \mathcal{A}$, and we have that $\tau(s_{31}, a) = s_{31}$ for both $a \in \mathcal{A}$. At $s_0$, we have that $\tau(s_0, up) = s_1$ and $\tau(s_0, down) = s_2$, and at $s_1$, we have that $\tau(s_1, a)$ for both $a \in \mathcal{A}$ returns $s_0$ with probability 0.99, and otherwise returns $s_{31}$. The reward function $R$ is zero everywhere, except that $R(s_{30}, a, s_{31}) = 100$ for both $a \in \mathcal{A}$, and $R(s_0, up, s_1) = 1$. The discount $d$ is the hyperbolic discount function, $d(t) = 1/(1+t)$. This environment is depicted in the following graph:*

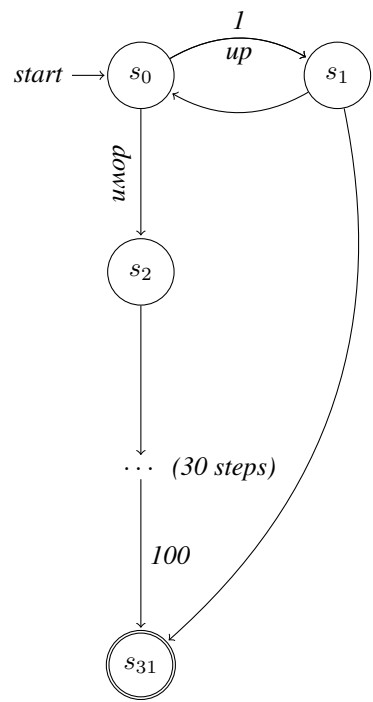

*Note that* Tempt *is episodic, with $s_{31}$ being the terminal state. Moreover, state $s_0$ is the only state in which the agent has a meaningful choice to make; in all other states, $\tau$ does not depend on the action. Note also that $\tau$ is deterministic everywhere, except at $s_1$ – the nondeterminism at $s_1$ is to ensure that* Tempt *is episodic.*

**Proposition 7.** *There exists episodic MDPs in which every sophisticated policy is nondeterministic.*

*Proof.* Consider the MDP Tempt, given in Example 3, and let $\pi$ be any deterministic policy. There are now two cases; either $\pi$ always selects up, or there exists a $\xi$ such that $\pi(\xi) = $ down.

Case 1: Suppose $\pi(\xi) = $ up for all $\xi$. We then have

$$Q^\pi(\xi, \text{up}) \approx 3.008 \quad Q^\pi(\xi, \text{down}) = 3.\overline{3}$$

We thus have that $Q^\pi(\xi, \text{down}) > Q^\pi(\xi, \text{up})$, even though $\pi(\xi) = $ up. This means that $\pi$ is not sophisticated.

Case 2: Suppose $\pi(\xi) = $ down for some $\xi$. We then have

$$Q^\pi(\xi, \text{up}) \approx 4.125 \quad Q^\pi(\xi, \text{down}) = 3.\overline{3}$$

We thus have that $Q^\pi(\xi, \text{up}) > Q^\pi(\xi, \text{down})$, even though $\pi(\xi) = $ down. This means that $\pi$ is not sophisticated.

Since Case 1 and 2 are exhaustive, this means that no deterministic policy is sophisticated in Tempt. However, Tempt is episodic, so by Theorem 4, there must be a policy that is sophisticated in Tempt. Hence, every sophisticated policy in Tempt is nondeterministic. $\square$

**Proposition 8.** *There exists an episodic MDP $M$ and policies $\pi_1, \pi_2$ such that both $\pi_1$ and $\pi_2$ are sophisticated in $M$, but $Q^{\pi_1} \neq Q^{\pi_2}$.*

*Proof.* Consider the MDP `Tempt` $= \langle \mathcal{S}, \mathcal{A}, \tau, \mu_0, R, d \rangle$, given in Example 3, and let `Tempt`$_2 = \langle \mathcal{S}, \mathcal{A}, \tau, \mu_0, R_2, d \rangle$ be the MDP that is identical to `Tempt`, except that $R_2 = -R$. Let $\pi_{\text{up}}$ be the policy that always chooses the action up, and $\pi_{\text{down}}$ be the policy that always chooses the action down. We now have that $Q^{\pi_{\text{up}}}$ and $Q^{\pi_{\text{down}}}$ are given by:

$$Q^{\pi_{\text{up}}}(s_0, \text{up}) \approx -3.008 \qquad Q^{\pi_{\text{up}}}(s_0, \text{down}) = -3.\overline{3}$$
$$Q^{\pi_{\text{down}}}(s_0, \text{up}) \approx -4.125 \qquad Q^{\pi_{\text{down}}}(s_0, \text{down}) = -3.\overline{3}$$

From this, we have that both $\pi_{\text{up}}$ and $\pi_{\text{down}}$ are sophisticated. However, $Q^{\pi_{\text{up}}} \neq Q^{\pi_{\text{down}}}$. $\qquad \square$

### A.8 IDENTIFIABILITY

**Theorem 5.** *Assume we have an episodic MDP, let $u(t) = 1$, and let $\pi_1$ and $\pi_2$ be policies such that*
$$\mathcal{J}_u(\pi_1) > \mathcal{J}_u(\pi_2).$$
*Then if $h(t) = 1/(1 + k \cdot t)$, then there exist an $N \in \mathbb{N}$ such that for all $n \geq N$, if $h^{+n}(t) = h(t+n)$, we have*
$$\mathcal{J}_{h^{+n}}(\pi_1) > \mathcal{J}_{h^{+n}}(\pi_2).$$
*Moreover, there is a $\Gamma \in (0, 1)$ such that, for all $\gamma \in [\Gamma, 1)$, if $e^\gamma(t) = \gamma^t$, then we have that*
$$\mathcal{J}_{e^\gamma}(\pi_1) > \mathcal{J}_{e^\gamma}(\pi_2).$$

*Proof.* We will prove this by showing that
$$\lim_{n \to \infty} (1 + kn)\mathcal{J}_{h^{+n}}(\pi) = \lim_{\gamma \to 1} \mathcal{J}_{e^\gamma}(\pi) = \mathcal{J}_u(\pi).$$

From this, it follows that if $\mathcal{J}_u(\pi_1) > \mathcal{J}_u(\pi_2)$, then $\mathcal{J}_{h^{+n}}(\pi_1) > \mathcal{J}_{h^{+n}}(\pi_2)$ and $\mathcal{J}_{e^\gamma}(\pi_1) > \mathcal{J}_{e^\gamma}(\pi_2)$ for all sufficiently large $n$, and all $\gamma$ sufficiently close to 1. Note that the $(1 + kn)$-term is a scaling term included to prevent $\mathcal{J}_{h^{+n}}(\pi)$ from approaching zero – the precise purpose of this will be made more clear later.

Recall that if $\lim_{x \to \infty} f_i(x)$ exists, and if $\sum_{i=0}^{\infty} f_i$ converges uniformly, then

$$\lim_{x \to \infty} \sum_{i=0}^{\infty} f_i(x) = \sum_{i=0}^{\infty} \lim_{x \to \infty} f_i(x).$$

Recall also that a sequence of functions $\sum_{i=0}^{\infty} f_i$ converges uniformly if for all $\epsilon$ there is a $J$ such that if $j \geq J$ then $|\sum_{i=0}^{j} f_i(x) - \sum_{i=0}^{J} f_i(x)| \leq \epsilon$ for all $x$.

We first apply this to hyperbolical discounting. Let

$$f_i(n) = \left( \frac{1 + kn}{1 + k(n+i)} \right) \mathbb{E}_\pi [R_i].$$

That is, $f_i(n)$ is the expected reward of $\pi$ at the $i$'th step, discounted as though it were the $(n + i)$'th step using hyperbolic discounting with parameter $k$, and rescaled such that the first step is not discounted (i.e. so that it is multiplied by 1). Now $(1 + kn)\mathcal{J}_{h^{+n}}(\pi) = \sum_{i=0}^{\infty} f_i(n)$.

We can begin by noting that $\lim_{n \to \infty} f_i(n)$ exists, and that it is equal to $\mathbb{E}_\pi [R_i]$. To show that $\sum_{i=0}^{\infty} f_i$ converges uniformly, recall that Lemma 2 says that there exists a $t$ and a $p$ such that for any policy $\pi$ and any state $s$, we have that if $\pi$ is run from $s$, then it will after $t$ steps have entered a terminal state with probability at least $p$. Moreover, since $\mathcal{S}$ and $\mathcal{A}$ are finite, we have that $m = \max_{s,a,s'} |R(s, a, s')| < \infty$. This means that $|\mathbb{E}_\pi [R_i]| \leq mp^{\lfloor i/t \rfloor}$, which in turn also means that $|f_i(n)| \leq mp^{\lfloor i/t \rfloor}$, since $(1 + kn)/(1 + k(n+i)) \in [0, 1]$. This implies that for all $\ell$,

$$\left| \sum_{i=\ell \cdot t}^{\infty} f_i(n) \right| \leq \frac{mtp^\ell}{1 - p}.$$

By making $\ell$ large enough, this quantity can be made arbitrarily close to 0. Thus $\sum_{i=0}^{\infty} f_i$ converges uniformly. We therefore have that

$$
\begin{aligned}
\lim_{n \to \infty} (1 + kn)\mathcal{J}_{h+n}(\pi) &= \lim_{n \to \infty} \sum_{i=0}^{\infty} f_i(n) \\
&= \sum_{i=0}^{\infty} \lim_{n \to \infty} f_i(n) \\
&= \sum_{i=0}^{\infty} \mathbb{E}_\pi [R_i] \\
&= \mathcal{J}_c(\pi)
\end{aligned}
$$

Thus, if we have that $\mathcal{J}_c(\pi_1) > \mathcal{J}_c(\pi_2)$, then it follows that $\lim_{n \to \infty}(1 + kn)\mathcal{J}_{h+n}(\pi_1) > \lim_{n \to \infty}(1 + kn)\mathcal{J}_{h+n}(\pi_2)$. Moreover, we of course have that $\mathcal{J}_{h+n}(\pi_1) > \mathcal{J}_{h+n}(\pi_2)$ if and only if $(1 + kn)\mathcal{J}_{h+n}(\pi_1) > (1 + kn)\mathcal{J}_{h+n}(\pi_2)$. Thus $\lim_{n \to \infty} \mathcal{J}_{h+n}(\pi_1) > \lim_{n \to \infty} \mathcal{J}_{h+n}(\pi_2)$, which in turn means that there exist an $N \in \mathbb{N}$ such that for all $n \geq N$, we have $\mathcal{J}_{h+n}(\pi_1) > \mathcal{J}_{h+n}(\pi_2)$. This completes the first part.

For the second part, simply let

$$
f_i(\gamma) = \gamma^i \mathbb{E}_\pi [R_i].
$$

That is, $f_i(\gamma)$ is the expected reward of $\pi$ at the $i$'th step, exponentially discounted with discount factor $\gamma$. Now $\mathcal{J}_{e^\gamma}(\pi) = \sum_{i=0}^{\infty} f_i(\gamma)$. We of course have that $\lim_{\gamma \to 1} f_i(\gamma)$ exists, and that it is equal to $\mathbb{E}_\pi [R_i]$, and we can show that $\sum_{i=0}^{\infty} f_i$ converges uniformly using the same argument as before. We therefore have that

$$
\begin{aligned}
\lim_{\gamma \to 1} \mathcal{J}_{e^\gamma}(\pi) &= \lim_{\gamma \to 1} \sum_{i=0}^{\infty} f_i(\gamma) \\
&= \sum_{i=0}^{\infty} \lim_{1 \to \gamma} f_i(\gamma) \\
&= \sum_{i=0}^{\infty} \mathbb{E}_\pi [R_i] \\
&= \mathcal{J}_c(\pi)
\end{aligned}
$$

Thus, if $\mathcal{J}_c(\pi_1) > \mathcal{J}_c(\pi_2)$, then $\lim_{\gamma \to 1} \mathcal{J}_{e^\gamma}(\pi_1) > \lim_{\gamma \to 1} \mathcal{J}_{e^\gamma}(\pi_2)$, which in turn means that there is a $\Gamma \in (0, 1)$ such that, for all $\gamma \in [\Gamma, 1)$, we have that $\mathcal{J}_{e^\gamma}(\pi_1) > \mathcal{J}_{e^\gamma}(\pi_2)$. This completes the second part, and the proof. $\square$

**Theorem 6.** *Let $d$ be a discount function, and let $f_{\tau,d}$ be a behavioural model that is regularly resolute, regularly naïve, or regularly sophisticated, for transition function $\tau$ and discount $d$. Then for any $\gamma \in (0, 1]$, unless there is an $\alpha \in (0, 1]$ such that $d(t) = \alpha\gamma^t$ for all $t \leq |\mathcal{S}| - 2$, there exists a transition function $\tau$ such that $f_\tau$ is not $\mathrm{OPT}_{\tau,\gamma}$-identifiable.*

*Proof.* Pick an arbitrary discount function $d$ and exponential discount rate $\gamma$, and assume that there is no $\alpha$ such that $d(t) = \alpha\gamma^t$ for all $t \leq |\mathcal{S}| - 2$.

First assign an integer value to every state in $\mathcal{S}$, so that $\mathcal{S} = \{s_0 \ldots s_n\}$, where $s_0 \in \mathrm{supp}(\mu_0)$ and $s_n$ is the terminal state. We assume that $\mathcal{A}$ contains at least two actions $a_1, a_2$. Now consider the transition function $\tau$ where $\tau(s_0, a_1) = s_1$ and $\tau(s_0, a_i) = s_n$ for all $a_i \neq a_1$. For $i \in \{1 \ldots n-1\}$, let $\tau(s_i, a) = s_{i+1}$ for all $a$, and let $\tau(s_n, a) = s_n$ for all $a$. This function can be visualised as:

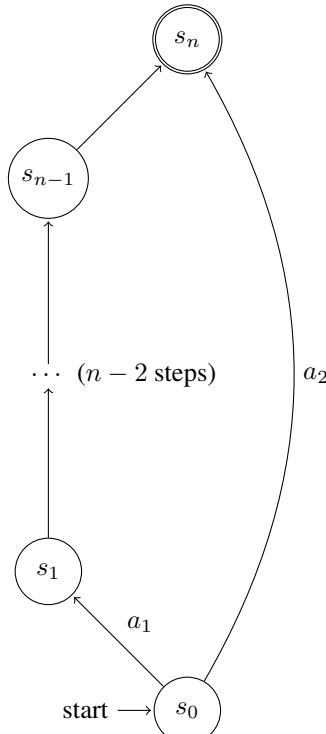

Let the reward function $R$ be selected arbitrarily, and let $\langle \mathcal{S}, \mathcal{A}, \tau, \mu_0, R, d \rangle$ be the resulting MDP. Let $\alpha = d(0)$. By assumption, there is no $\alpha$ such that $d(t) = \alpha\gamma^t$ for all $t \le |\mathcal{S}| - 2$, and so there must be a $t \le |\mathcal{S}| - 2$ such that $d(t) \ne \alpha\gamma^t$. From the construction of $\alpha$, we also have that it must be the case that $t \ne 0$.

Let $R_1$ be selected arbitrarily, and consider the reward function $R_2$ where $R_2(s_0, a, s_n) = R_1(s_0, a, s_n) + x/d(0)$ for all $a \ne a_1$, $R_2(s_t, a, s_{t+1}) = R_1(s_t, a, s_{t+1}) + x/d(t)$ for all $a$, and $R' = R$ for all other transitions.[7] We now have that $R_1$ and $R_2$ share the same resolute and naïve advantage function, i.e. $A_1^{\mathrm{R}} = A_2^{\mathrm{R}}$ and $A_1^{\mathrm{N}} = A_2^{\mathrm{N}}$. Moreover, for any policy $\pi$, we have that and $A_1^\pi = A_2^\pi$. Therefore, since $f_{\tau,d}$ is regularly resolute, regularly naïve, or regularly sophisticated, we have that $f_{\tau,d}(R_1) = f_{\tau,d}(R_2)$.

However, since $d(t) \ne d(0)\gamma^t$, we can ensure that $R_1$ and $R_2$ have different optimal policies (under discounting with $\gamma$), by making $x$ sufficiently large or sufficiently small. To see this, note that $Q_2^\star(s_0, a_1) - Q_1^\star(s_0, a_1) = x \cdot \gamma^t/d(t)$, and $Q_2^\star(s_0, a_i) - Q_1^\star(s_0, a_i) = x/\alpha$ for $a_i \ne a_1$. Since $d(t) \ne \alpha\gamma^t$, these quantities are not equal. Thus, if $a_1$ is an optimal action at $s_0$ under $R_1$ and $\gamma^t/d(t) > 1/\alpha$, then for any $x$ that is sufficiently negative, we have that $a_1$ is not an optimal action at $s_0$ under $R_2$. Similarly, if $a_1$ is an optimal action at $s_0$ under $R_1$ and $\gamma^t/d(t) < 1/\alpha$, then $x$ has to be sufficiently *large*, and so on. We can therefore always ensure that $R_1$ and $R_2$ have different optimal actions at $s_0$.

Thus, for all $R_1$ there is an $R_2$ such that $f_{\tau,d}(R_1) = f_{\tau,d}(R_2)$, but $R_1$ and $R_2$ have different optimal policies. Thus $f$ is not $\mathrm{OPT}_{\tau,\gamma}$-identifiable. $\qquad\square$

**Theorem 7.** *Let $d$ be a discount function, let $\tau$ be a non-trivial acyclic transition function, and let $f_{\tau,d}$ be a behavioural model that is regularly resolute, regularly naïve, or regularly sophisticated, for transition function $\tau$ and discount $d$. Then for any $\gamma \in (0, 1]$, unless $\gamma = d(1)/d(0)$, we have that $f_{\tau,d}$ is not $\mathrm{OPT}_{\tau,\gamma}$-identifiable.*

*Proof.* Let $\tau$ be an arbitrary non-trivial acyclic transition function, let $\gamma \in (0, 1]$ be selected arbitrarily, and let $d$ be an arbitrary discount function such that $\gamma \ne d(1)/d(0)$. Moreover, let $R_1$ be an

---

[7]In other words, if the agent goes right at $s_0$, it will immediately receive an extra $x/d(0)$ reward, and if it goes left, it will receive an extra $x/d(t)$ reward after $t$ steps.

arbitrary reward function. We will show that there exists a reward function $R_2$ such that $R_1$ and $R_2$ have different optimal policies (under $\tau$ and $\gamma$), but $f_{\tau,d}(R_1) = f_{\tau,d}(R_2)$.

Recall that a state $s'$ is *controllable* if there is a non-terminal state $s$ and actions $a_1, a_2$ such that $\mathbb{P}(\tau(s, a_1) = s') \neq \mathbb{P}(\tau(s, a_2) = s')$. Since $\tau$ is non-trivial, there is at least one controllable state. Moreover, since $\tau$ is acyclic, and since $\mathcal{S}$ is finite, there must be a controllable state that cannot be reached from any other controllable state. Call this state $s_c$. Since $s_c$ is not terminal, there are states which are reachable from $s_c$.

Now let $R_2$ be the reward function where $R_2(s, a, s_c) = R_1(s, a, s_c) + x/d(0)$ and $R_2(s_c, a, s) = R_1(s_c, a, s) - x/d(1)$ for all $s$ and $a$, and $R_2 = R_1$ for all other transitions. We now have that $R_1$ and $R_2$ share the same resolute and naïve advantage function, i.e. $A_1^{\mathrm{R}} = A_2^{\mathrm{R}}$ and $A_1^{\mathrm{N}} = A_2^{\mathrm{N}}$. Moreover, for any policy $\pi$, we have that and $A_1^{\pi} = A_2^{\pi}$. To see this, note that:

1. In all states $s$ which are neither reachable from $s_c$, nor able to reach $s_c$, we of course have that $A_1^{\{*\}} = A_2^{\{*\}}$, for $* \in \{\mathrm{R}, \mathrm{N}, \pi\}$. $R_1$ and $R_2$ only differ on transitions that begin or end in $s_c$, and so they must induce the same advantage functions in states which are disconnected from $s_c$.

2. In all states $s$ which are reachable from $s_c$, we also have that $A_1^{\{*\}} = A_2^{\{*\}}$, for $* \in \{\mathrm{R}, \mathrm{N}, \pi\}$. Again, $R_1$ and $R_2$ only differ on transitions that begin or end in $s_c$. Since $\tau$ is acyclic, we have that if a state $s$ is reachable from $s_c$, then it cannot reach $s_c$. Thus $R_1$ and $R_2$ must induce the same advantage functions in such states.

3. In $s_c$, we have that every outgoing transition gets an extra $x \cdot d(0)/d(1)$ reward, and that any subsequent transition after that is unchanged. This straightforwardly means that for all actions $a$, we have that $Q_2^{\mathrm{N}}(s_c, a) = Q_1^{\mathrm{N}}(s_c, a) + x \cdot d(0)/d(1)$, and that $Q_2^{\pi}(s_c, a) = Q_1^{\pi}(s_c, a) + x \cdot d(0)/d(1)$ for all $\pi$. Thus $A_2^{\mathrm{N}}(s_c, a) = A_1^{\mathrm{N}}(s_c, a)$ and $A_2^{\pi}(s_c, a) = A_1^{\pi}(s_c, a)$. Similarly, $A_2^{\mathrm{R}}(s_c, t, a) = A_1^{\mathrm{R}}(s_c, t, a)$ for all $t$.

4. Finally, for the most complicated case, suppose $s$ can reach $s_c$, and let $a$ be an arbitrary action. Let $A_{s,a}$ be the difference between the expected future discounted $R_1$-reward and $R_2$-reward, if you take action $a$ in state $s$ and then following $\pi$, conditional on the event that $\tau(s, a)$ returns a state which is controllable from $s$. Moreover, let $B_{s,a}$ be the difference between the expected future discounted $R_1$-reward and $R_2$-reward, if you take action $a$ in state $s$ and then following $\pi$, conditional on the event that $\tau(s, a)$ returns a state which is *not* controllable from $s$. Now $Q_2^{\pi}(s, a) = Q_1^{\pi}(s, a) + A_{s,a} + B_{s,a}$. Moreover, from the definition of controllable states, we have that $B_{s,a_1} = B_{s,a_2}$ for all actions $s_1, s_2$, and so we can express this variable as $B_s$. Next, note that if a state $s'$ is controllable from $s$, then either $s' = s_c$, or $s_c$ is not reachable from $s'$ (since $s_c$ is not reachable from any controllable state). If $s' \neq s_c$, and $s_c$ is not reachable from $s'$, then the difference in future discounted $R_1$-reward and $R_2$-reward, conditional on transitioning to $s'$, is zero. Similarly, the difference in future discounted $R_1$-reward and $R_2$-reward, conditional on transitioning to $s_c$, is $d(0) \cdot x/d(0) - d(1) \cdot x/d(1) = 0$. Thus, $A_{s,a} = 0$, and each $Q$-function is shifted by a constant value $B_s$, which means that the advantage functions are unaffected.

Thus $R_1$ and $R_2$ share the same resolute and naïve advantage function, i.e. $A_1^{\mathrm{R}} = A_2^{\mathrm{R}}$ and $A_1^{\mathrm{N}} = A_2^{\mathrm{N}}$. Moreover, for any policy $\pi$, we have that and $A_1^{\pi} = A_2^{\pi}$. Therefore, since $f_{\tau,d}$ is regularly resolute, regularly naïve, or regularly sophisticated, we have that $f_{\tau,d}(R_1) = f_{\tau,d}(R_2)$.

However, by making $x$ sufficiently large or sufficiently small, we can ensure that $R_1$ and $R_2$ have different optimal policies. To see this, note that since $s_c$ is controllable, there must be a state $s_i$ and actions $a_1, a_2$ such that $\mathbb{P}(\tau(s_i, a_1) = s_c) \neq \mathbb{P}(\tau(s_i, a_2) = s_c)$. Let $\mathbb{P}(\tau(s_i, a_1) = s_c) = p$ and $\mathbb{P}(\tau(s_i, a_2) = s_c) = q$. Since $\tau$ is acyclic, we have that $Q_2^{\star}(s, a) = Q_1^{\star}(s, a)$ for all states $s$ which are reachable from $s_c$, and $Q_2^{\star}(s_c, a) = Q_1^{\star}(s_c, a) - x/d(1)$ for all $a$. However, in $s_i$, we have that $Q_2^{\star}(s_i, a_1) = Q_1^{\star}(s_i, a_1) + p(x/d(0) - \gamma x/d(1))$ and $Q_2^{\star}(s_i, a_2) = Q_1^{\star}(s_i, a_2) + q(x/d(0) - \gamma x/d(1))$. Since $\gamma \neq d(1)/d(0)$, we have that $x/d(0) - \gamma x/d(1) \neq 0$. Moreover, $p \neq q$. Therefore, by making $x$ larger or smaller, we can increase the value of $Q_2^{\star}(s_i, a_1)$ relative to $Q_2^{\star}(s_i, a_2)$, and vice versa. In particular, if $Q_1^{\star}(s_i, a_1) \geq Q_1^{\star}(s_i, a_2)$, then we can ensure that $Q_1^{\star}(s_i, a_1) < Q_1^{\star}(s_i, a_2)$, and vice versa. This means that we can ensure that $R_1$ and $R_2$ have

different optimal policies. Thus, for all $R_1$ there is an $R_2$ such that $f_{\tau,d}(R_1) = f_{\tau,d}(R_2)$, but $R_1$ and $R_2$ have different optimal policies. Thus $f_{\tau,d}$ is not $\mathrm{OPT}_{\tau,\gamma}$-identifiable. $\qquad\square$

We should also note that Theorem 7 will be hard to generalise, without adding assumptions. To see this, consider a transition function that looks as follows:

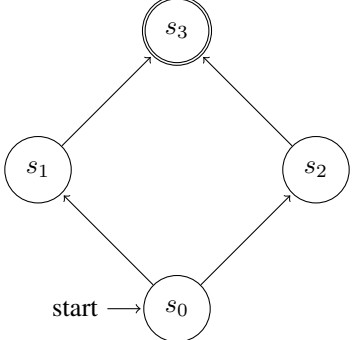

This transition function is acylclic and non-trivial, but here, for any $\gamma$ and any discount function $d$ such that $d(1)/d(0)$, we have that any regularly resolute, regularly naïve, or regularly sophisticated behavioural model $f_{\tau,d}$ is $\mathrm{OPT}_{\tau,\gamma}$-identifiable. This makes it tricky to generalise Theorem 7, without adding stronger assumptions about $d$.