# OpenReview forum: "Characterising Partial Identifiability in Inverse Reinforcement Learning For Agents With Non-Exponential Discounting"
_ICLR.cc/2024/Conference — Submitted to ICLR 2024_

### Official Review · Reviewer_HKSo · 2023-10-26

**Soundness:** 2 fair
**Presentation:** 3 good
**Contribution:** 2 fair
**Rating:** 5
**Confidence:** 2

**Summary:**

This paper explores the challenge of partial identifiability in Inverse Reinforcement Learning (IRL) under the condition of non-exponential discounting. Specifically, it focuses on a hyperbolic discounting model, which is characterized by temporal inconsistency, thereby inducing non-stationarity in the underlying Markov Decision Process (MDP).

To address the effects of this temporal inconsistency, the paper proposes a series of behavioral models: the resolute policy, the naive policy, and the sophisticated policy. For each of these models, the paper succinctly summarizes their properties, encompassing the uniqueness of the optimal value function, the stochastic nature of the policy, and the stationarity of the policies across varying time steps.

Interestingly, the paper defines the identifiability of reward functions in relation to the optimal policy under an exponential discounting setting. This appears contradictory to the paper's main focus on non-exponential discounting.

The theoretical findings indicate that no regularly resolute, regularly naive, or regularly sophisticated behavioral model is identifiable under non-exponential discounting or a non-trivial acyclic transition function. These results suggest that IRL is incapable of inferring sufficient information about rewards to identify the correct optimal policy. Consequently, it is implied that IRL alone is insufficient to thoroughly characterize the preferences of such agents.

**Strengths:**

1. The matter of identifiability in Inverse Reinforcement Learning (IRL) under a non-exponential discounting setting has yet to be explored in prior studies.

2. The paper's overall structure is logically organized and easily navigable. Definitions are meticulously presented, supplemented with numerous intuitive examples to facilitate reader understanding of the core content.

3. The theoretical findings are well presented, thereby supporting the claims made in the paper.

**Weaknesses:**

1. It's challenging to comprehend the concept of sophisticated policy as delineated in Definition 7. For instance, it's unclear why the policy, $\pi(\xi)$, is not dependent on the time step and how it correlates with step-wise policies. Similarly, it's puzzling why the Q function $Q^\pi(\xi,a)$ is also independent of the time step. Given that the optimal policy can vary at each time step, it becomes complex to determine which strategy exhibits more "sophistication". In many Markov Decision Processes (MDPs), the so-called sophisticated policy is not singular. The paper states that "$\pi$ is sophisticated if it only takes actions that are optimal given that all subsequent actions are sampled from $\pi$." Could you clarify this definition? Specifically, I'm interested in understanding how one would define optimality in a non-stationary MDP that spans across different (or all) time steps.

2. The definition of identifiability appears to be founded on an exponentially discounted MDP, even though the paper focuses on a non-exponentially discounted setting. The paper attempts to provide some intuitive explanations, but they fall short in terms of persuasiveness. If the term 'optimality' has a clear definition under different behavior models, then the term 'identifiability' should also exhibit the capacity to characterize these models.

3. This paper lacks empirical studies to substantiate its arguments. The main results suggest that IRL alone may be inadequate to fully characterize the preferences of agents in a non-exponentially discounted setting. However, a potential solution has not been proposed, and it is yet unclear how the existence of non-identifiability impacts empirical performance. It would be beneficial to see these points addressed in future research.

**Questions:**

1. why the policy, $\pi(\xi)$, is not dependent on the time step and how it correlates with step-wise policies?
2. why the Q function $Q^\pi(\xi,a)$ is also independent of the time step ?
3. How to understand the "sophisticated policy"?
4. how the existence of non-identifiability impacts empirical performance?
5. What potential solutions could address the issue of non-identifiability in IRL?

---

> ### Author Response · Authors · 2023-11-13
> **Response**
>
> We thank reviewer HKSo for their review! Our responses to your questions are as follows:
>
> Weaknesses:
> 1. To say that a policy is sophisticated is essentially to say that it satisfies a certain stability criterion --- namely, if $\pi$ is sophisticated, then the agent never has any (local) incentive to deviate from $\pi$. One way to understand this is that a sophisticated policy is somewhat analogous to a Nash equilibrium, if the agent at each time step is thought of as a separate decision maker (which is justified by the fact that its preferences are not temporally consistent). Another way to understand it is that if a policy $\pi$ is updated using policy gradients, then a sophisticated policy will be a sort of local optimum (though it may not necessarily be an attractor point, in the same way as a Nash equilibrium may not be an attractor point). This also means that it is not always clear how to compare sophisticated policies (again, in the same way as how it is not always clear how to select between Nash equilibria); a policy is simply either sophisticated or not. Thus, sophisticated policies do not give us an unambiguous notion of "optimality" (though they are nonetheless a solution concept).
> 2. We have choosen to quantify identifiability in terms of exponentially discounted optimal policies, because even if humans discount hyperbolically, we typically want to create systems that discount exponentially --- this is why exponential discounting is vastly more common in the RL literature. Therefore, we think the most relevant formalisation is to assume that the observed demonstrator discounts hyperbolically, but that the policy which will be computed with the learnt reward function discounts exponentially. Other choices could be made instead, which might lead to different results. Of course, some alternative formalisations would end up being quite trivial. For example, if $P$ is the equivalence relation under which $R_1 \equiv_P R_2$ iff $R_1$ and $R_2$ have the same sophisticated policies, and $f : \mathcal{R} \to \Pi$ returns a maximally supportive sophisticated policy, then it is immediate from the definition that $f$ is $P$-identifiable, etc. This setup would thus not allow for the derivation of any "deep" results.
> 3. We agree that experimental evaluations could be interesting, but we consider this to be out of scope for this paper. Similar recent works on identifiability in IRL are also typically theoretical, rather than experimental (see Dvijotham & Todorov, 2010; Cao et al., 2021; Kim et al., 2021; Skalse et al., 2022; Schlaginhaufen & Kamgarpour, 2023; Metelli et al., 2023). Note that the empirical performance of an IRL algorithm for hyperbolic discounting will be heavily dependent on the prior distribution from which the ground truth reward function is sampled, which limits the generalisability of such results. In particular, for any inductive bias of the learning algorithm there exists a prior distribution for the ground truth reward such that the counterexamples identified by our theorems will occur with high probability.
>
> Questions:
> 1. Note that $\xi$ is a trajectory --- this means that $\pi(\xi)$ *does* depend on the time step, since it may be sensitive to the length of $\xi$.
> 2. Again, note that $\xi$ is a trajectory, rather than a state.
> 3. See above.
> 4. See above.
> 5. We think that the most promising solutions will rely on incorporating prior information, or data from other data sources. However, exploring these options is beyond the scope of this paper.
>
> We hope that this clarifies our analysis, and that the reviewer might consider increasing their score!

---

> > ### Comment · Reviewer_HKSo · 2023-11-19
> >
> > I am still struggling with the term "identifiability" under the non-stationary policy and MDP. Since all the policies are locally optimal, are the rewards also locally identifiable? or since we consider an exponentially discounted MDP when we define the term "identifiability", we can utilize the corresponding time consistency to define globally identifiable rewards.
> >
> > As for the assumption "assume that the observed demonstrator discounts hyperbolically, but that the policy which will be computed with the learnt reward function discounts exponentially.". I don't think such an assumption is intuitive and well justified, especially when it utilizes time-consistent policies (under exponential discounting) to explain the time-inconsistent (under hyperbolical discounting) behaviors. Without empirical results, it is hard to evaluate the effect of such an assumption.

---

> > > ### Author Response · Authors · 2023-11-19
> > >
> > > > Since all the policies are locally optimal, are the rewards also locally identifiable?
> > >
> > > What do you mean by "locally identifiable" in this context? We cannot determine the exact value of the reward for any transition in any state. We can also in general not determine whether or not $R(s,a_1) > R(s,a_2)$ for any $s$, $a_1$ or $a_2$, etc.
> > >
> > > > since we consider an exponentially discounted MDP when we define the term "identifiability", we can utilize the corresponding time consistency to define globally identifiable rewards.
> > >
> > > What do you mean by "define globally identifiable rewards"? We cannot uniquely determine the reward function of the demonstrator. This is true both in this setting, and in the exponentially discounted setting.
> > >
> > > > I don't think such an assumption is intuitive and well justified.
> > >
> > > Humans, and many other animals, are better modelled as using hyperbolic discounting rather than exponential discounting. For this reason, we think it is justified and relevant to consider the case where the demonstrator is assumed to use hyperbolic discounting. Similarly, we often want policies that discount exponentially, for the reasons outlined in Section 5. Intuitively speaking, such policies are more patient, and this is usually desirable. More generally, in IRL, we typically want to take demonstrations from a sub-optimal demonstrator and use these to compute a better policy. One way to view this paper is that it considers a particular form of systematic sub-optimality that the demonstrator may exhibit (namely hyperbolic discounting), and whether or not this sub-optimality can be accounted for and cancelled out.
> > >
> > > > especially when it utilizes time-consistent policies (under exponential discounting) to explain the time-inconsistent (under hyperbolical discounting) behaviors.
> > >
> > > Our apologies, but we are again not quite sure what you mean here. We study a setup where time-inconsistent behaviour in the demonstrator is explained by a combination of the discount function and the reward function that the demonstrator uses. In particular, we assume that the demonstrator uses non-exponential discounting, and we attempt to find a reward function that explains the behaviour of the demonstrator. This is exactly the usual setting for IRL, except that we assume that the demonstrator discounts non-exponentially, rather than exponentially. We are then attempting to characterise how accurately we are able to identify the reward function of the demonstrator in this setting. We are not assuming that the IRL algorithm uses a misspecified behavioural model. Could you please clarify this point?

---

### Official Review · Reviewer_nfpA · 2023-10-31

**Soundness:** 2 fair
**Presentation:** 2 fair
**Contribution:** 2 fair
**Rating:** 3
**Confidence:** 3

**Summary:**

The paper defines novel MDP concepts based on novel definitions of discount factors. The authors started by presenting in the background the standard exponential discounting setting. Then they define the non-exponential setting, defining in section 4 the optimality conditions for the policies. Finally, they studied when behavioral models for inverse reinforcement learning are identifiable.

**Strengths:**

- The paper provides novel results on partial identifiability in IRL with non-exponential discounting.

- They provide the first theoretical results on IRL in a non-exponential discounting setting.

**Weaknesses:**

- The main weakness of the work is the motivation of it. The authors do not provide enough reasons why we need to consider a different discounted setting with respect to the exponential discounted one. In literature, when is it used the hyperbolic setting? Why is it relevant in practice? Moreover, If the setting is more general and reasonable, I think it would be better to present directly it in the background section rather than presenting the standard exponential discounted ones and then the new setting.

- The main focus of the paper is (reading the abstract) on Inverse Reinforcement Learning, but, in the end, the IRL contribution of the paper is condensed into only one page and a half.

- There are no experimental or numerical evaluations of the proposed approach at least to show why the proposed setting is relevant.

**Questions:**

- A reward function is optimal under more than one policy. Then, why is the behavioral model defined as a mapping between $\mathcal{R} \rightarrow \Pi$ and not $\mathcal{R} \rightarrow P^\Pi$?

- Proposition 1 seems to be not easy to verify. How can we understand if an MDP satisfies it?

- Why is it relevant to choose discount factors that are not temporally consistent? Can the change in preference of an agent be described with a change in the reward function?

- If in the end, we are using exponential discounting to find our optimal policy why do we need to study a different setting before?

---

> ### Author Response · Authors · 2023-11-13
> **Response**
>
> First of all, we would like to thank reviewer nfpA for their thoughts and questions. Our responses to your feedback and questions are as follows:
>
> Weaknesses:
> 1. First of all, we would like to note that the main motivation behind our work is the usage of IRL in the context of *preference elicitation*. That is, the setting where we wish to use IRL to learn a representation of the actual preferences of a human subject. In this setting, it is important that we use a behavioural model that actually represents the decision making process of a human as closely as possible. Now, a vast amount of research in behavioural psychology, and related fields, shows that humans are better modelled as using hyperbolic discounting rather than exponential discounting --- we provide references to some of this research in the introduction to the paper. This means that a behavioural model with hyperbolic discounting will be more accurate than (and hence preferable to) a behavioural model with exponential discounting, all other things being equal. Note also that this is not necessarily the case when IRL is used in the context of *imitation learning*, since it in this context is not fundamentally important that the learnt reward function represents the actual preferences of the demonstrator, as long as it helps the imitation learning process.
> 2. Yes, the main results on IRL are given in Section 5, but all results in Section 3 and 4 build towards these results. It would not be possible to derive (or even state) the results in Section 5 without first presenting the results given in Section 3 and 4.
> 3. We agree that experimental evaluations could be interesting, but we consider this to be out of scope for this paper. Similar recent works on identifiability in IRL are also typically theoretical, rather than experimental (see Dvijotham & Todorov, 2010; Cao et al., 2021; Kim et al., 2021; Skalse et al., 2022; Schlaginhaufen & Kamgarpour, 2023; Metelli et al., 2023).
>
> Questions:
> 1. This choice was made to make our analysis more tractable. For example, if an optimality criterion corresponds to more than one policy, then we may assume that the demonstrator has some fixed criterion for breaking ties between them. This assumption is not fundamentally important, and can be lifted, at the cost of making some of the theorem statements and proofs more messy. For example, both Theorem 6 and 7 would go through without much modification.
> 2. To say that an MDP is episodic is simply to say that it cannot run forever; most MDPs used in practice satisfy this criterion. For example, note that any MDP with a bounded time horizon is episodic. The definition used in Proposition 1 is simply somewhat more general.
> 3. As noted before, the current literature in the behavioural sciences suggests that humans are better modelled as using hyperbolic discounting, rather than exponential discounting. For this reason, we consider the setting with non-exponential discounting in IRL to be both very relevant and under-explored.
> 4. To quantify the "size" of $\mathrm{Am}(f)$, we need to determine if all reward functions in $\mathrm{Am}(f)$ share some relevant property. There are multiple choices that could be made here, but we have choosen to use (exponentially discounted) optimal policies, for the reasons outlined in Section 5. In short, even if humans discount hyperbolically, we typically want to create systems that discount exponentially --- this is why exponential discounting is vastly more common in the RL literature. Therefore, we should assume that the observed demonstrator discounts hyperbolically, but that the policy which will be computed with the learnt reward function discounts exponentially.
>
> We hope that this clarifies the context of our paper, and that the reviewer will consider increasing their score! We believe that this setting is very relevant, and that our results meaningfully extend the existing literature in interesting and non-trivial ways.

---

### Official Review · Reviewer_wg8J · 2023-11-04

**Soundness:** 3 good
**Presentation:** 3 good
**Contribution:** 2 fair
**Rating:** 5
**Confidence:** 2

**Summary:**

This paper studies the partial identifiability problem in IRL with non-exponential discounting; the authors provide their theoretical conclusion that for some behavioral models with non-exponential discounting, the partial identifiability problem persists.

**Strengths:**

There are a few theoretical results that seems quite interesting and potentially significant. I appreciate the clear definitions and background. However, I am unable to determine whether these results are easily ported results or more original findings.

**Weaknesses:**

So much of the proof is deferred to the appendix, it would be helpful if a proof sketch is summarized in the main text.

**Questions:**

From R we could get to different f(R), which is denoted Am(f), a set of rational models follows R. Rather than knowing this set is singleton, I think a more important question maybe how small the set is, and whether it is contiguous. Do you think non-exponential discounting effects contiguity?

---

> ### Author Response · Authors · 2023-11-13
> **Response**
>
> We thank reviewer wg8J for their feedback!
>
> As for the first point, our results and proofs are original and non-trivial extensions of the existing literature. Only a small handful of papers have studied non-exponential discounting in IRL, and none of them have considered the (quite important) problem of partial identifiability. Concepts analogous to sophisticated, naïve, and resolute policies have been proposed in the Decision Theory literature, but doing the work of porting these over to the RL setting is non-trivial, and has (to the best of our knowledge) not been done before. As you can see from our proofs, it is not straightforward to show that these policies are guaranteed to exist in all MDPs, etc. The results about identifiability (in Section 5) are completely original.
>
> The proofs take up ~10 pages, so it would difficult to let the main text include even proof sketches for all our results. However, we can move the MDP construction used in the proof of Theorem 6 to the main text, since this proof is quite central and since the MDP construction is fairly illuminating. We can also add a few short remarks to other proofs, such as e.g. "we can use the Kakutani fixed-point theorem to show that...", etc.
>
> As for the last point, we would expect $\mathrm{Am}(f)$ to be contiguous in the non-exponential setting, but we do not have a proof of this fact. We should also note that we agree that it is not very important whether or not $\mathrm{Am}(f)$ is a singleton. Indeed, in both the exponential and the non-exponential setting, $\mathrm{Am}(f)$ is infinitely large (both in the sense that it has an infinite cardinality, and in the sense that it has an infinite Lebesgue measure). Therefore, we think it is more important to know whether or not all reward functions in $\mathrm{Am}(f)$ are "equivalent" in some relevant sense. This is why we use the notion of $P$-identifiability to express our results in Section 5.
>
> We hope that this clarifies our results and contributions, and that the reviewer may consider increasing their score!

---

### Meta-Review · Area_Chair_PQSo · 2023-12-06

**Metareview:**

a) Claims: The paper considers the setting in which demonstrations have been produced by an agent that discounts in a potentially time-inconsistent way (e.g., hyperbolically) rather than exponentially.  It proves that such policies do not provide enough information for inverse RL to infer the optimal policy with respect to the true R.

b) Strengths: Most reviewers agreed that the results were clearly organized.  IRL is an important area, and this paper is the first to explore the specific question of identifiability under alternative time-inconsistent discounting schemes.

c) Weaknesses: It wasn't clear to some of the reviewers how relevant the specific claims were.  Multiple reviewers commented on the lack of empirical support for the claims; this partly led to the ambivalence about the significance of the results.  The proofs were completely deferred to the appendix; it is good practice to at least try to give some intuition or a proof sketch for the most important results.

**Justification For Why Not Higher Score:**

The paper has a lot going for it, but the reviewers were unanimous that it does not meet the bar for publication.  The paper should make a more compelling argument for why it is solving the right problem; for instance, that the optimality definition is the right one, and/or empirical checks to demonstrate that the targeted problems can actually arise.

**Justification For Why Not Lower Score:**

n/a

---

### Decision · Program_Chairs · 2024-01-16

Reject